# QIL1 is a novel mitochondrial protein required for MICOS complex stability and cristae morphology

Virginia Guarani[1], Elizabeth M McNeill[1], Joao A Paulo[1], Edward L Huttlin[1], Florian Fröhlich[1,2], Steven P Gygi[1], David Van Vactor[1], J Wade Harper[1]*

[1]Department of Cell Biology, Harvard Medical School, Boston, United States; [2]Department of Genetics and Complex Diseases, Harvard T.H. Chan School of Public Health, Boston, United States

**Abstract** The mitochondrial contact site and cristae junction (CJ) organizing system (MICOS) dynamically regulate mitochondrial membrane architecture. Through systematic proteomic analysis of human MICOS, we identified QIL1 (C19orf70) as a novel conserved MICOS subunit. QIL1 depletion disrupted CJ structure in cultured human cells and in *Drosophila* muscle and neuronal cells in vivo. In human cells, mitochondrial disruption correlated with impaired respiration. Moreover, increased mitochondrial fragmentation was observed upon QIL1 depletion in flies. Using quantitative proteomics, we show that loss of QIL1 resulted in MICOS disassembly with the accumulation of a MIC60-MIC19-MIC25 sub-complex and degradation of MIC10, MIC26, and MIC27. Additionally, we demonstrated that in QIL1-depleted cells, overexpressed MIC10 fails to significantly restore its interaction with other MICOS subunits and SAMM50. Collectively, our work uncovers a previously unrecognized subunit of the MICOS complex, necessary for CJ integrity, cristae morphology, and mitochondrial function and provides a resource for further analysis of MICOS architecture.

*For correspondence:
wade_harper@hms.harvard.edu

Competing interests: The authors declare that no competing interests exist.

## Introduction

Mitochondria exhibit a complex topology encompassing two membranes that create distinct internal compartments. The inner membrane (IM) runs parallel to the outer membrane (OM) at regions called inner boundary membranes (IBM) and invaginates into the mitochondrial matrix forming the cristae structures which connect to the inter membrane space (IMS) through narrow openings (*Freya and Mannellab, 2000*). This intricate architecture is maintained by structures called cristae junctions (CJs) and contact sites (CSs) and has been shown to be essential for numerous mitochondrial pathways such as protein import, oxidative phosphorylation, and apoptosis (*Freya and Mannellab, 2000*; *Vogel et al., 2006*; *Yang et al., 2012*; *Cogliati et al., 2013*).

It has been reported that changes in physiological energetic states and oxygen availability affect the morphology of cristae and CJs (*Lorente et al., 2002*; *Walker and Benzer, 2004*), suggesting that the formation of CJs is a dynamic process that promotes cristae remodeling as an adaptive mechanism to the different needs and metabolic states of the cell. Moreover, several mitochondrial disorders in humans are often accompanied by alterations of mitochondrial ultrastructure including MERRF syndrome (Myoclonic epilepsy with ragged red fibers), BTHS (Barth syndrome), fatal neonatal lactic acidosis, mtDNA defects and hypertrophic cardiomyopathy and loss of CJs with the formation of concentric stacks of cristae membrane are some of the hallmarks observed in mitochondria from such patients (*Silva-Oropeza et al., 2004*; *Acehan et al., 2007*; *Roels et al., 2009*; *Götz et al., 2012*). As such, deciphering how CJs and CSs are formed and regulated is crucial for the understanding of cristae dynamics and maintenance in health and disease.

**eLife digest** Mitochondria are the cell's power plants, and churn out molecules that provide a portable energy source throughout the cell. To do this efficiently, the mitochondria have a double membrane. The inner membrane is ruffled, which provides a large surface area for energy-producing reactions to occur on. Structures called cristae junctions and contact sites hold the folds of the inner membrane in place.

As mitochondria are found in every cell in the body, mitochondrial diseases can produce a wide range of symptoms, but they commonly affect the muscles. In some forms of these diseases, the inner membrane of a mitochondrion is no longer folded; instead, the membrane may form concentric rings like the layers of an onion. Knowing how the folding of the inner membrane is regulated may therefore help scientists to better understand mitochondrial diseases.

Scientists already know that several proteins join together to form a complex that anchors the mitochondrion's inner membrane to its outer membrane at cristae junctions. To learn more about the proteins involved in these complexes, Guarani et al. systematically screened for proteins that associate with cristae junctions and found a previously unknown protein called QIL1.

Next, Guarani et al. conducted a series of experiments to determine what role QIL1 plays at the cristae junctions. The experiments showed that QIL1 is needed to bind a protein called MIC10 into the protein complex that anchors the cristae junctions to the outer membrane. In human and fruit fly cells without QIL1, this protein complex falls apart and is not repaired if extra MIC10 is added into the cells. Furthermore, in human cells lacking QIL1, the inner mitochondrial membrane forms the same onion-like rings seen in the cells of humans with mitochondrial diseases. Future studies are necessary to understand how the structure of the QIL1 complex is organized and to work out how the complex is capable of causing the mitochondrial inner membrane to curve.

Despite its critical importance, only recently have components of a protein complex that localizes at CJs and CSs and are responsible for mitochondrial cristae architecture been identified (*van der Laan et al., 2012*). Recent studies in both yeast and human cells led to the identification of various subunits of a highly conserved heterooligomeric protein complex of approximately ~700 kDa, referred to as the mitochondrial contact site and cristae junction organizing system (MICOS), which controls the formation of CJs and CSs and IM morphology (*Gieffers et al., 1997*; *John et al., 2005*; *Xie et al., 2007*; *Rabl et al., 2009*; *Darshi et al., 2011*; *Harner et al., 2011*; *Hoppins et al., 2011*; *von der Malsburg et al., 2011*; *Alkhaja et al., 2012*; *An et al., 2012*; *Bohnert et al., 2012*; *Jans et al., 2013*; *Korner et al., 2012*; *Ott et al., 2012*; *Pfanner et al., 2014*; *Weber, 2013*). A recent study has proposed to unify the nomenclature for MICOS, with subunits of this complex termed MIC10 to MIC60 (with subunit mass indicated by the numbers) (*Pfanner et al., 2014*). To date, 7 bona fide subunits have been described in yeast (Mic10, Mic12, Mic19, Mic25, Mic26, Mic27, Mic60), 6 of which have obvious human orthologs (MIC10/MINOS1, MIC19/CHCHD3, MIC25/CHCHD6, MIC27/APOOL, MIC26/APOO and MIC60/Mitofilin) (*Icho et al., 1994*; *Odgren et al., 1996*; *Gieffers et al., 1997*; *John et al., 2005*; *Xie et al., 2007*; *Rabl et al., 2009*; *Mun et al., 2010*; *Darshi et al., 2011*; *Harner et al., 2011*; *Head et al., 2011*; *Hoppins et al., 2011*; *von der Malsburg et al., 2011*; *Alkhaja et al., 2012*; *An et al., 2012*; *Bohnert et al., 2012*; *Korner et al., 2012*; *Ott et al., 2012*; *Itoh et al., 2013*; *Jans et al., 2013*; *Weber, 2013*). The MIC10-MIC19-MIC25-MIC26-MIC27-MIC60 complex is thought to reside in the IM at the site of CJs. Genetic removal of several MICOS subunits causes mitochondrial fragmentation, reduced respiration, CJs loss, and the formation of concentric stacks of IM disconnected from the IBM (*John et al., 2005*; *Rabl et al., 2009*; *Darshi et al., 2011*; *Harner et al., 2011*; *Hoppins et al., 2011*; *Alkhaja et al., 2012*; *Ott et al., 2012*; *van der Laan et al., 2012*; *Weber, 2013*). These phenotypes strikingly resemble cristae abnormalities observed in patients with the aforementioned mitochondrial disorders. Moreover, alterations of several MICOS subunit protein levels, mutations and protein modifications have been linked to a variety of human diseases, including diabetic cardiomyopathy, epilepsy, Parkinson's disease, and cancer (*Zerbes et al., 2012*). One study has employed proteomic analysis of MIC10/MINOS1 protein interactors in human cells to identify further components of the complex, and in addition to the IM

core subunits, this work also identified the OM proteins SAMM50, MTX1, MTX2, and the chaperone DNAJC11 (*Alkhaja et al., 2012*). A role for these proteins (*Xie et al., 2007*) and in particular the sorting and assembly machinery protein SAMM50 in CJ assembly was also found independently (*Ott et al., 2012*), suggesting a role for β-barrel insertion into the OM in CJ formation or maintenance. Interestingly, mutations in DNAJC11 in the mouse result in mitochondrial disruption, cristae abnormalities, and motor neuron pathology (*Ioakeimidis et al., 2014*).

Given the crucial role that the MICOS complex plays in CJ architecture and mitochondrial homeostasis (*Zerbes et al., 2012*), a further understanding of its constituents and regulatory mechanisms is needed. To develop a comprehensive understanding of MICOS composition and regulation, we performed proteomic IP-MS analysis of the MICOS complex. In addition to core MICOS subunits, we identified several new candidate MICOS-associated proteins. One of these—QIL1—is a previously unstudied IM protein conserved from human to *Drosophila*. Using quantitative proteomics coupled with native gel analysis, we show that QIL1 is a component of a ~700 KDa MICOS complex and its depletion from cells results in loss of MIC10, MIC26, and MIC27 from the complex and a reduction in the abundance of these proteins in mitochondria. Thus, QIL1 appears to be required for incorporation of MIC10, MIC26, and MIC27 into the MICOS complex. Depletion of QIL1 in human cells results in several mitochondrial phenotypes including loss of CJs, cristae rearrangement into stacks of concentric membranes, and reduced respiration. In *Drosophila* muscle and neuronal cells, QIL1 depletion leads to analogous morphological phenotypes. This study provides the most comprehensive protein interaction network map of the MICOS complex to date and will serve as a resource for further elucidation of MICOS assembly and regulation.

## Results

### Systematic proteomic analysis of the MICOS complex

In order to examine the composition of the MICOS complex using interaction proteomics, open reading frames for MIC27, MIC19, MIC25, MIC60, MTX2, and DNAJC11 (*Figure 1—figure supplement 1A* shows a schematic representation of the MICOS complex where the MICOS subunits and interactors used for our IP-MS approach are depicted in red) were C-terminally tagged with an HA-FLAG epitope in a lentiviral vector and expressed stably in 293T and HeLa cells (*Figure 1—figure supplement 1B*). Confocal microscopy after immunostaining with α-HA and α-TOMM20 verified that each protein was targeted to mitochondria in HeLa cells (*Figure 1A*). To identify high confidence interacting proteins (HCIPs), we employed a modified version of the *ComPASS* platform (*Sowa et al., 2009*). This method uses a large collection of parallel AP-MS experiments to generate a database populated with peptide spectral matches, allowing the frequency, abundance, and reproducibility of interacting proteins to be determined. To enhance detection of membrane-associated proteins, we employed 1% digitonin, and proteins were purified using α-FLAG beads. After extensive washing, complexes were trypsinized prior to proteomic analysis. As a validation approach, three of the baits (MIC60, MTX2, and MIC19) were also expressed in HCT116 cells and immunopurified with a different antibody (α-HA). Interaction data are summarized in *Figure 1B* (*Figure 1—figure supplement 2* contains the entire data set). Overall, the interaction network contained 26 proteins and 97 interactions (edges) after filtering as described in the 'Materials and methods'. The six baits analyzed showed extensive reciprocal connectivity (*Figure 1B*). Confirming previously reported data, several core subunits of the MICOS complex (MIC19, MIC25, MIC60, MIC26, MIC27) also associated with known interactors at the OM (SAMM50, MTX1 and MTX2), indicating that our method is able to retrieve nearly all known subunits and interactors of the MICOS complex, located at the IM, IMS, and OM with high confidence. In addition to known interactors, our map also revealed potential novel interacting partners, associated with one or more MICOS subunits. These include two OM proteins, the MUL1 E3 ubiquitin ligase and the RHOT2 GTPase involved in mitochondrial trafficking (*Figure 1B*). RHOT2 has been shown to co-fractionate with SAMM50 in correlation profiling proteomic experiments (*Havugimana et al., 2012*). In addition, we identified TMEM11 as a protein associated with multiple MICOS subunits and capable of associating with MIC60 endogenously (*Figure 1B,F*). A TMEM11 ortholog in *Drosophila* has been shown genetically to be required for cristae organization and biogenesis, but the mechanisms involved are unknown (*Rival et al., 2011*; *Macchi et al., 2013*). Our results indicate that TMEM11 may function in these processes in association with the MICOS

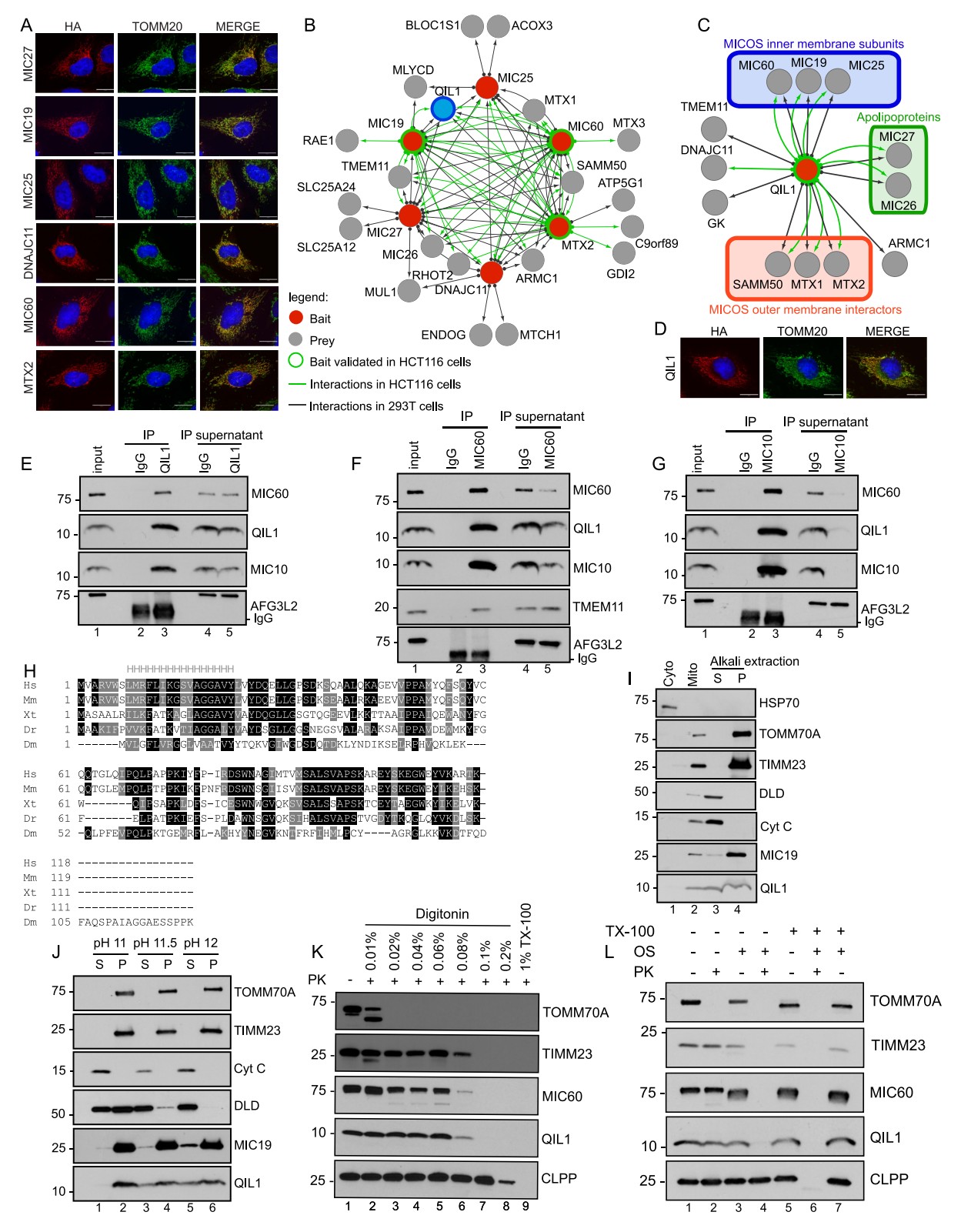

**Figure 1**. Interaction proteomics of the MICOS complex reveals QIL1 as a novel interactor. (**A**) Immunofluorescence analysis of the subcellular localization of tagged proteins. Bars, 20 μm. (**B**) Overview of the MICOS interaction network obtained from IP-MS analysis. (**C**) Validation of QIL1 protein interactions by IP-MS analysis. *Figure 1—figure supplement 1* shows a schematic representation of the MICOS complex highlighting the subunits and interactors

*Figure 1. continued on next page*

*Figure 1. Continued*

analyzed by IP-MS in red and the expression levels of C-terminally tagged proteins compared to the endogenous version. *Figure 1—figure supplement 2* contains the entire IP-MS data set. (**D**) Immunofluorescence analysis of the subcellular localization of C-terminally tagged QIL1. Bars, 20 μm. Endogenous QIL1 (**E**), MIC60 (**F**), or MIC10 (**G**) was immunopurified from crude mitochondria isolated from 293T cells. Endogenous interactions with MIC60, MIC10, AFG3L2, TMEM11, or QIL1 were assessed. (**H**) Alignment of QIL1 orthologs using the ClustalW software. 'H' indicates amino acids present within a conserved predicted transmembrane region (TMPred algorithm). (**I**) Cytoplasmic and mitochondrial fractions were separated in 293T lysates. Soluble and membrane fractions were separated by alkaline extraction. S indicates soluble, P indicates pellet. (**J**) Alkaline extraction was performed at increasing pH (pH11, pH11.5, pH12). (**K**) Proteinase K (PK) and increasing concentrations of digitonin were used to assess QIL1 sub-mitochondrial localization. (**L**) Proteinase K (PK), osmotic shock (OS), and Triton X-100 (TX100) were used to assess QIL1 sub-mitochondrial localization.

The following figure supplements are available for figure 1:

**Figure supplement 1**. Expression levels of C-terminally tagged MICOS subunits and related proteins analyzed by IP-MS in this study compared to the endogenous version.

**Figure supplement 2**. Mass spectrometry analysis of immunopurified protein interactors for QIL1, MIC27, MIC19, MIC25, MIC60, MTX2, DNAJC11 and GFP in 293T and/or HCT116 cells.

complex. Components of the MICOS complex were not detected in GFP-FLAG immune complexes prepared similarly (*Figure 1—figure supplement 2*), pointing the specificity of the interactions observed.

## Identification of QIL1 as a novel MICOS interacting protein

Our attention was drawn to a previously uncharacterized protein with unknown function—C19orf70 (also called QIL1)—which was detected in association with MIC19, MIC60, and MTX2 in both 293T FLAG IPs and HCT116 HA IPs and with MIC27 additionally in 293T (*Figure 1B*). As an initial approach for validating the interactions, C-terminally tagged QIL1 was subjected to IP-MS analysis. The result elicited the generation of an interaction map containing 13 nodes and 20 edges (interactions) (*Figure 1C*), wherein we identified 5 core MICOS subunits (MIC60, MIC19, MIC25, MIC26 and MIC27), 3 OM known interactors (SAMM50, MTX1 and MTX2) as well as the chaperone DNAJC11 and TMEM11 in association with QIL1. In a further attempt to validate these interactions, we first employed antibodies directed at endogenous QIL1 for immunoprecipitation (IP) and detected endogenous MIC60, but not the abundant mitochondrial IM protein AFG3L2 protein, after western blot analysis (*Figure 1E*). Reciprocally, endogenous QIL1 co-precipitated with immunopurified endogenous MIC60 (*Figure 1F*).

The transmembrane protein MIC10 was the only known MICOS subunit that was not detected by our proteomics approach, possibly due to its small size (78 residues) and predominantly hydrophobic peptides. To address if QIL1 also interacted with MIC10, we immunopurified endogenous MIC10 in isolated mitochondria from 293T cells and assessed binding to QIL1 by western blot analysis (*Figure 1G*). In addition to detecting MIC60 in MIC10 immune complex (*Alkhaja et al., 2012*), we also detected QIL1. Thus, QIL1 endogenously associates with components of the MICOS complex.

## QIL1 is an inner membrane-associated protein

We then examined the sub-cellular localization of QIL1. First, QIL1-HA-FLAG was found to co-localize with endogenous TOMM20 in HeLa cells, (*Figure 1D*). Second, endogenous QIL1 was detected in purified mitochondria from 293T cells by western blotting, with a level of enrichment similar to that found with other mitochondrial proteins (*Figure 1I*), confirming that QIL1 is a mitochondrial protein. Our bioinformatics analysis identified QIL1 orthologs across metazoans, including *Drosophila* (*Figure 1H*). Additionally, primary sequence analysis using the TMpred algorithm revealed that QIL1 contains one possible conserved transmembrane helix (*Figure 1H*). To investigate whether QIL1 is a transmembrane protein, we performed carbonate extraction on mitochondria isolated from 293T cells. As expected, the transmembrane proteins TOMM70A and

TIMM23 were resistant to alkaline extraction regardless of the increase in pH (*Figure 1I,J*). Most bona fide MICOS subunits described so far are transmembrane proteins, with the exception of MIC19 and MIC25 (*Pfanner et al., 2014*). Interestingly, MIC19 has been shown to be myristoylated at the N-terminus allowing its docking onto SAMM50, which is embedded in the OM, and possesses a CHCH domain at its C-terminus that binds to MIC60, present in the IM (*Darshi et al., 2012*) (see *Figure 1—figure supplement 1A*). MIC19 was only partially extracted from the membrane fraction with increasing pH (*Figure 1J*). Similarly, while entirely present in the membrane fraction at lower pH, QIL1 was partially extracted from the membrane fraction at higher pH (*Figure 1I,J*). We did not identify a predicted myristoylation site or a CHCH domain in QIL1. These results suggest that QIL1 may indeed possess membrane anchoring activity, being possibly anchored by a single conserved membrane spanning helix as predicted by the TMPred algorithm (*Hofmann and Stoffel, 1993*).

To examine further the topology of QIL1 in the IM, we employed digitonin permeabilization of isolated mitochondria combined with proteinase K digestion (*Figure 1K*) as previously described (*Sancak et al., 2013*). Mitochondria were isolated from HCT116 cells and incubated with increasing concentrations of digitonin in the presence of proteinase K (PK). Subsequently, samples were analyzed by immunoblotting for the OM protein TOMM70A, IM proteins TIMM23 and MIC60, and matrix protein CLPP (*Figure 1K*). These experiments showed that the topology of QIL1 is similar to the topology of the IM proteins TIMM23 and MIC60 (*Figure 1K*). To confirm the topology of QIL1, we performed Proteinase K digestion with and without osmotic shock in the presence or absence of Triton X-100 (*Figure 1L*) as described previously (*Harner et al., 2014*). QIL1 displayed resistance to Proteinase K treatment in isolated mitochondria, but became accessible to protease digestion when the OM was disrupted by osmotic shock (*Figure 1L*), as also found with the IM proteins TIMM23 and MIC60 (*Figure 1L*). Importantly, the matrix protein CLPP was only accessible to Proteinase K digestion when mitochondria were solubilized with 1% Triton X-100, demonstrating the integrity of the IM. Thus, we concluded that QIL1 behaves like an IM protein.

## QIL1 is enriched in CJs

QIL1 interaction partners suggested an association with CJs. To examine QIL1 localization directly, we employed immunogold labeling followed by transmission electron microscopy analysis (*Figure 2A–C*) and measured the distance between each gold particle and the nearest CJ in nanometers as described previously (*Jans et al., 2013*). QIL1-HA was predominantly located within 50 nm of CJs (*Figure 2A*), being therefore enriched at CJs to a similar degree as MIC25-HA (*Figure 2B*), used as a positive control. In contrast, NDUFA13-HA, a Complex I subunit, was distributed along cristae membranes (*Figure 2C*).

## QIL1 is a constituent of the mature MICOS complex

The enrichment of QIL1 at CJs and its physical interaction with multiple MICOS subunits strongly supports the notion that QIL1 is a subunit of the MICOS complex. Thus, we asked whether QIL1 is present in the ~700 kDa MICOS complex found in mitochondria from human cells (*Ott et al., 2012*). To address this question, mitochondria isolated from 293T cells stably expressing C-terminally tagged QIL1, MIC60, MIC19, or MIC25 (*Figure 1—figure supplement 1B*) were lysed in 1% digitonin and subjected to BN-PAGE followed by immunoblot analysis (*Figure 2D*). QIL1 was found to be predominantly localized at ~700 kDa (asterisks). Similarly, MIC60, MIC19, and MIC25 were found in a ~700 kDa complex as expected. Additionally, MIC60, MIC19, and MIC25 were also present in a smaller complex of ~500 kDa (two asterisks) suggesting the existence of an assembly intermediate or sub-complex containing these three proteins. Using 2-dimensional BN-PAGE electrophoresis, we found that endogenous QIL1 was present within the mature MICOS complex migrating at ~700 kDa (*Figure 2E*). Confirming our findings in one-dimensional BN-PAGE immunoblotting, MIC60, MIC19, and MIC25 were found in the ~700 kDa complex (asterisk) and in the smaller sub-complex (two asterisks), while MIC10 and QIL1 were primarily found in the ~700 kDa complex. To address whether QIL1 transiently associates with the dissociated MICOS subunits, we solubilized mitochondria with 1% Triton X-100, which leads to the dissociation of the MICOS complex into smaller complexes as previously described (*Harner et al., 2014*). While C-terminally

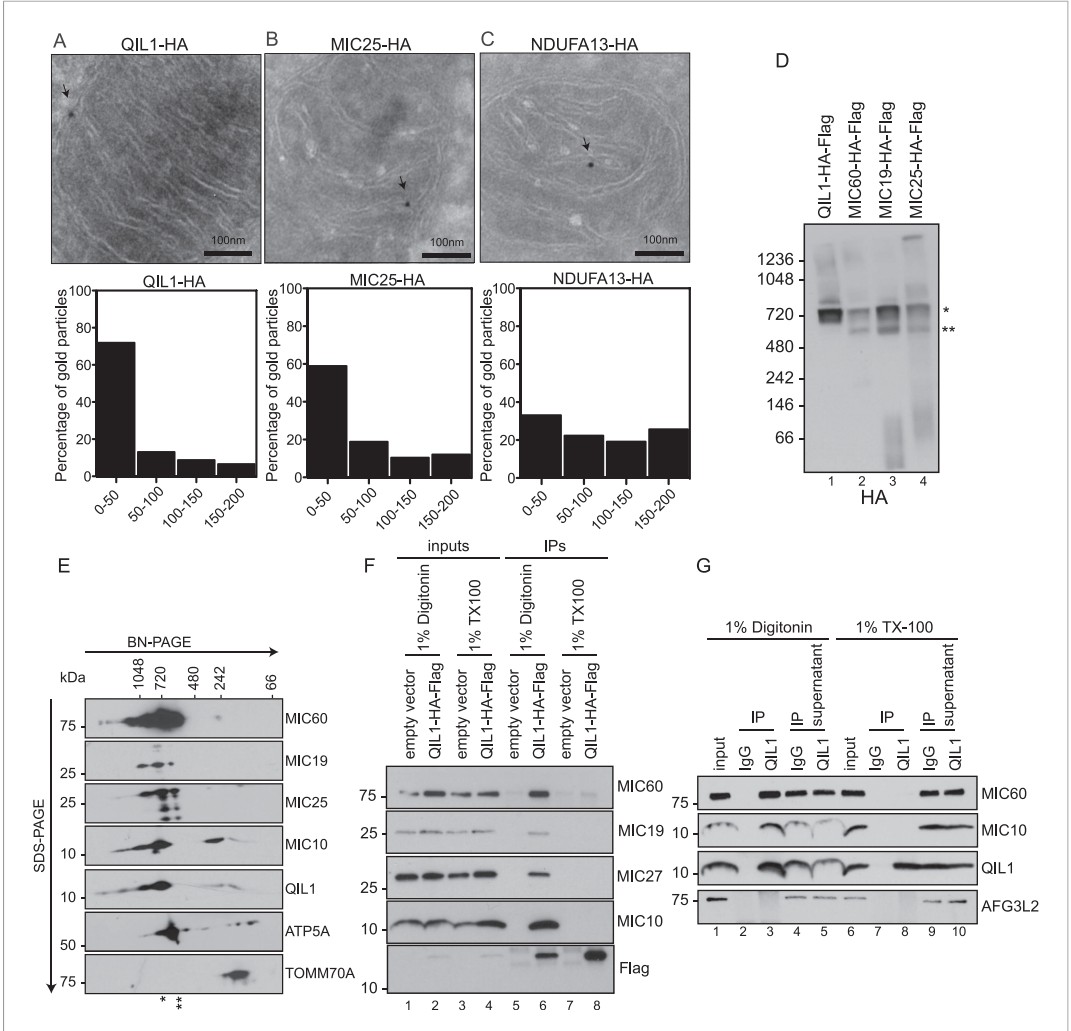

**Figure 2**. QIL1 localizes at cristae junctions and is present in the mature MICOS complex. (**A–C**) Immunogold labeling of 293T cells overexpressing C-terminally tagged QIL1 (**A**), MIC25 (**B**), or NDUFA13 (**C**) using an α-HA antibody coupled to 10-nm gold particles. Bars, 100 nm. Representative mitochondria are shown. The arrows point to the position of a gold particle. The distance in nanometers between the gold particles and the nearest CJ was measured using the ImageJ software. The histograms show the fraction of gold particles within the indicated distance to the crista junction in nanometers in QIL1-HA (n = 231 gold particles), MIC25-HA (n = 192 gold particles), and NDUFA13-HA (n = 309 gold particles) expressing cells. (**D**) Mitochondria isolated from stable 293T cell lines expressing C-terminally tagged QIL1, MIC60, MIC19, or MIC25 were lysed in 1% digitonin, subjected to BN-PAGE followed by immunotransfer to nitrocellulose membranes and probing with α-HA antibody. The mature ~700 kDa MICOS complex is highlighted with an asterisk. MIC60, MIC19, and MIC25 were also detected in a sub-complex of ~500 kDa (two asterisks). (**E**) Two-dimensional blue native electrophoresis of 293T mitochondrial lysates. Endogenous QIL1 is present in the mature ~700 kDa MICOS complex (asterisk). MIC60, MIC19, and MIC25 were also present in a smaller sub-complex (two asterisks). (**F–G**) C-terminally tagged (**F**) or endogenous QIL (**G**) was immunopurified from mitochondria lysed in 1% digitonin or 1% Triton X-100 (TX100). Immunoblot analysis was performed to detect interaction with other MICOS subunits.

tagged QIL1 efficiently co-purified with endogenous MIC60, MIC19, MIC27, and MIC10 when mitochondria were solubilized with 1% digitonin, no specific binding was detected when mitochondria were solubilized with Triton X-100, consistent with an association of QIL1 with the mature MICOS complex (*Figure 2F*). These findings were confirmed at the endogenous level recapitulating an interaction between QIL1 and MIC60, as well as MIC10, in mitochondria lysed with 1% digitonin but not with 1% Triton X-100 (*Figure 2G*).

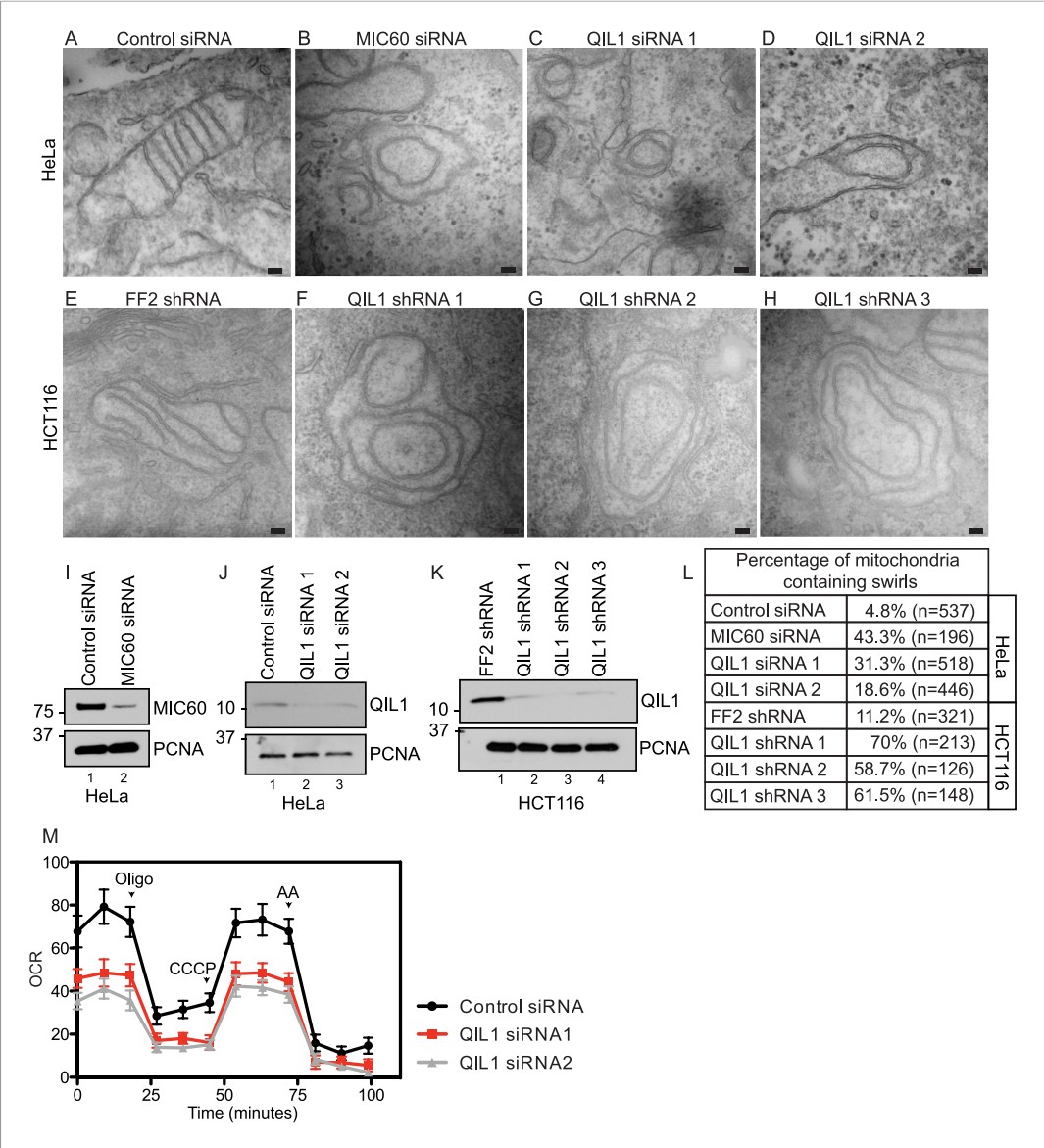

**Figure 3**. QIL1 deletion alters CJs formation, cristae morphology, and mitochondrial respiration. Electron microscopy of HeLa cells transfected with Control siRNA (**A**), MIC60 siRNA (**B**), two independent QIL1 siRNAs (**C** and **D**), FF2 shRNA (**E**), or QIL1 shRNAs (**F–H**). Bars, 100 nm. (**I–K**) Knock-down levels are shown by immunoblot analysis. (**L**) Quantification of the number of mitochondria containing membrane swirls based on electron microscopy images. (**M**) Oxygen consumption rate (pmoles/min) was measured at baseline conditions, after oligomycin, FCCP, and antimycin A injections as indicated by arrowheads in HeLa cells transfected with control or QIL1 siRNAs.

## QIL1 is required for the formation of CJs and maintenance of cristae morphology

We next examined mitochondrial cristae morphology upon RNAi-mediated QIL1 depletion using transmission electron microscopy (*Figure 3A–H*). In agreement with previously published data, transient depletion of MIC60 as a control (*Figure 3I*) resulted in dramatic changes in cristae organization. In particular, IM structures composed of one to several layers of concentric rings, resembling 'onion-like' structures, were found, with no connection to the IBM due to loss of CJs as reported previously (*John et al., 2005*) (*Figure 3B*). Similarly, QIL1 depletion in HeLa (*Figure 3J, C–D*) and HCT116 (*Figure 3K,F–H*) cells resulted in analogous rearrangement of cristae structures.

Quantification of electron microscopy images revealed a dramatic increase in the number of mitochondria containing swirls upon MIC60 or QIL1 depletion (*Figure 3L*).

As seen previously with other MICOS subunits (*Darshi et al., 2011*; *Weber, 2013*), depletion of QIL1 resulted in a substantial reduction in respiration, suggesting an important role for QIL1 in mitochondrial homeostasis through CJ formation (*Figure 3M*).

## *Drosophila* QIL1 regulates mitochondrial network and cristae morphology

Our bioinformatics analysis identified an apparent QIL1 ortholog (CG7603) in *Drosophila* (*Figure 1H*). Muscle tissue is rich in mitochondria and relies strongly on mitochondrial function. Using the UAS/Gal4 system, and specifically the Dmef2-Dcr-Gal4 driver to express UAS-Control[RNAi] or UAS-QIL1[RNAi], we achieved significant depletion of QIL1 mRNA in *Drosophila* third larval instar bodywall muscle (*Figure 4A*). To address whether QIL1 depletion resulted in the same ultrastructural defects in mitochondrial cristae morphology as those observed in human cell lines, we dissected muscles from crawling third instar larvae and submitted the tissues to transmission electron microscopy analysis (*Figure 4B–C*). Strikingly, upon QIL1 depletion, we observed a significant increase in the number of abnormal mitochondria, with many mitochondria showing loss of CJs and concentric stacks of IM inside the matrix compartment (*Figure 4C*). Quantification revealed an ~10-fold increase in the number of mitochondria containing IM swirls (*Figure 4D*). Moreover, we observed an increase in mitochondrial fragmentation and sphericity in the *Drosophila* muscle upon QIL1 depletion as shown by our confocal microscopy analysis (*Figure 4E–J*). Similarly, silencing of QIL1 specifically in neurons using the Elav-Dcr-Gal4 driver to express UAS-Control[RNAi] or UAS-QIL1[RNAi] (*Figure 4K*) led to loss of CJs and the formation of concentric stacks of IM inside the matrix compartment in neurons at neuromuscular junctions (*Figure 4L–M*). Quantification of the number of mitochondria containing IM swirls in neurons revealed a significant increase upon QIL1 knockdown (*Figure 4N*). Together, these results demonstrate that QIL1 loss destabilizes CJs and alters cristae morphology in different cell types in vivo in a cell autonomous fashion.

## Depletion of QIL1 impairs MICOS assembly

To gain insight into the molecular mechanism by which QIL1 causes morphological changes in cristae membrane organization, we coupled SILAC (stable isotopic labeling with amino acids in culture) with size-based fractionation of mitochondria purified from cells with or without QIL1 to investigate whether QIL1 is required for MICOS complex formation (*Figure 5A*). Briefly, SILAC-labeled cells with and without QIL1 depletion were mixed 1:1, mitochondria isolated, subjected to BN-PAGE, and gel slices across the entire mass range of the BN-PAGE were subjected to LC-MS[2]. We analyzed the relative abundance of all subunits of the MICOS complex, with the exception of MIC10, which was not quantified by mass spectrometry perhaps due to its small size and the unfavorable properties of its peptides following tryptic digestion as discussed previously. Quantitative proteomic analysis revealed a marked reduction of MICOS subunits at ~700 kDa (*Figure 5A*; H:L ratio <1, green), corresponding to the mature heterooligomeric complex (*Harner et al., 2011*; *Ott et al., 2012*) and concomitant accumulation of MIC19, MIC25, and MIC60 in a smaller ~500 kDa sub-complex and below in response to QIL1 depletion (*Figure 5A*; H:L ratio >1, red). Moreover, when we analyzed the estimated relative protein abundance across all fractions, the total protein abundance for MIC26 and MIC27 were reduced upon QIL1 depletion (*Figure 5B*).

To confirm these findings, we turned to western blot analysis. Mitochondria were isolated from HCT116 cells stably expressing FF2 shRNA or three different shRNAs-targeting QIL1. Mitochondria were lysed in 1% digitonin and subjected to BN-PAGE followed by immunoblot analysis (*Figure 5C*). Antibodies against MIC60, MIC19, and MIC10 were used to assess MICOS assembly. ATP5A antibody was used as a control, and QIL1 knockdown was confirmed by submitting an input sample to SDS-PAGE. Again, we observed a decrease in the abundance of the ~700 kDa complex with all subunits tested. Moreover, MIC10, which was not detected by mass spectrometry, was regulated in a similar fashion as the other MICOS subunits (*Figure 5C*). Interestingly, there was an accumulation of MIC60 and MIC19 at lower molecular weights (~500 kDa), as was also observed in the SILAC experiment for MIC60, MIC19, and MIC25, at ~500 kDa and below (*Figure 5C*), suggesting the accumulation of an assembly intermediate containing these proteins in the absence of QIL1.

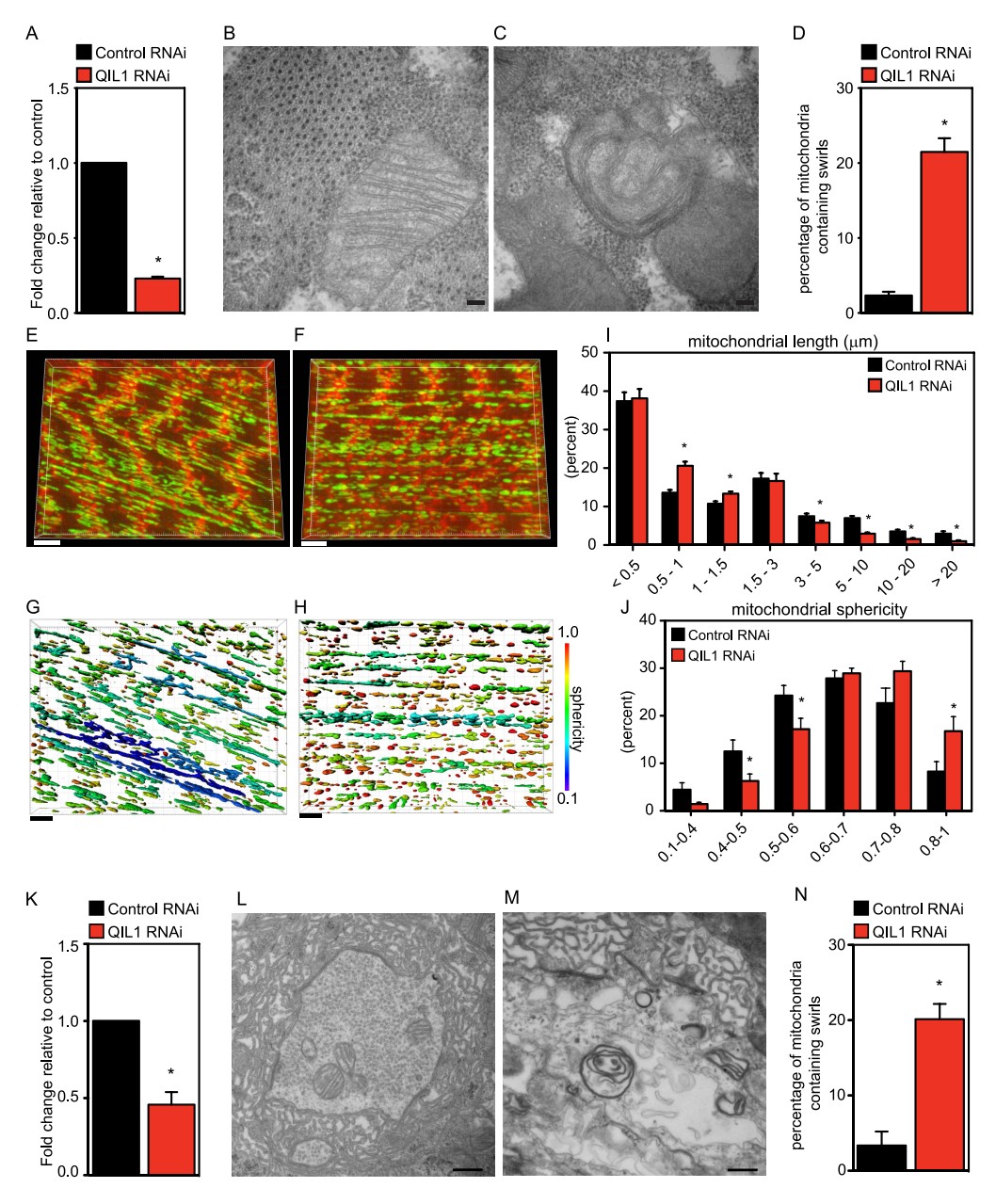

**Figure 4**. *Drosophila* QIL1 is required for mitochondrial homeostasis and CJ formation in vivo. (**A**) DMef2-driven QIL1 knock-down efficiency in three-instar larvae muscles was assessed by qPCR. Ultrastructural analysis of Control (**B**) vs QIL1 RNAi (**C**) third instar larval bodywall muscles in transversal sections. Bars, 100 nm. (**D**) The number of mitochondria containing IM swirls was quantified. Immunofluorescence analysis of muscles dissected from Control (**E**) or QIL1 RNAi (**F**) third instar larvae stained with Phalloidin (red) and α-ATP5A (green). Bars, 10 µm. (**I–J**) Mitochondrial length (**I**) and sphericity (**G-H**, **J**) of each individual mitochondria were measured. (**K**) Elav-driven QIL1 knock-down efficiency in three-instar larvae neurons was assessed by qPCR. Ultrastructural analysis of Control (**L**) vs QIL1 RNAi (**M**) third instar larval neuromuscular junctions in longitudinal sections. Bars, 500 nm. (**N**) The number of mitochondria containing IM swirls was quantified.

We analyzed protein levels for several MICOS subunits as well as the OM interactor, SAMM50, by SDS-PAGE (*Figure 5D*). This analysis revealed that, while most proteins remained unchanged after QIL1 knockdown, MIC27, MIC26, and MIC10 levels were significantly reduced (*Figure 5D–E*), confirming our quantitative proteomics experiment. Reduced levels of the MIC26, MIC27 and MIC10

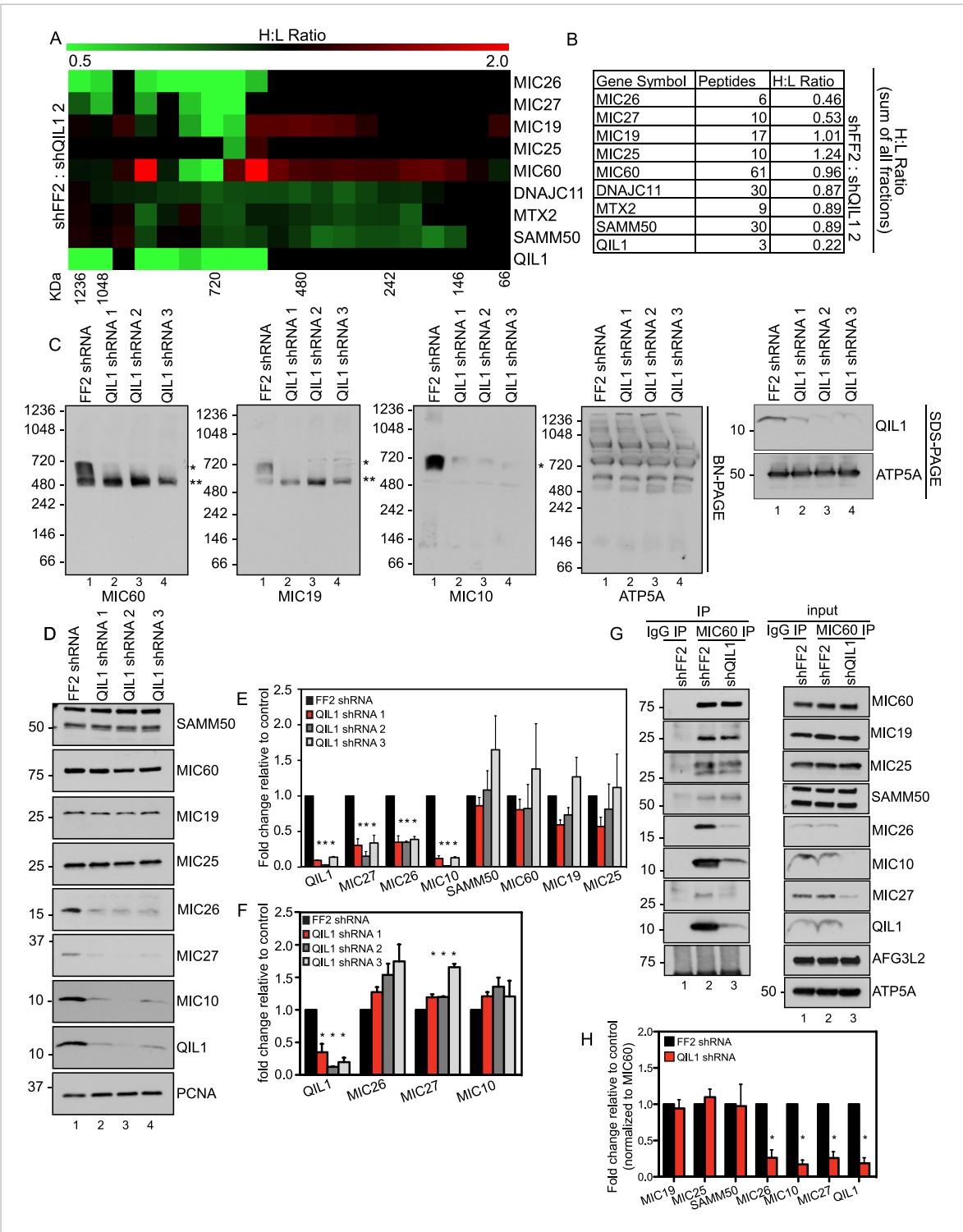

**Figure 5**. QIL1 deficiency impairs MICOS assembly. (**A**) Light-labeled (K0) shFF2 and heavy (K8)-labeled shQIL1 stable cell lines were mixed at a 1:1 ratio and mitochondria subsequently isolated and lysed with 1% Digitonin. Mitochondrial protein complexes were separated using BN-PAGE and sliced in 20 gel pieces ranging from >1 MDa to <60 kDa. Proteins were subjected to tryptic digestion and analyzed by quantitative mass spectrometry. Heavy to light (H:L) ratios were calculated for MICOS components. The ratios were plotted in heatmaps where values <1 are represented in green and values >1 are represented in red. (**B**) Ratios from the summed heavy and light intensities for each peptide separately across all BN-PAGE fractions from *Figure 4A*. (**C**) BN-PAGE followed by immunotransfer to nitrocellulose membranes. QIL1 knockdown lead to a decrease in MIC60, MIC19, and MIC10 in the ~700 kDa mature MICOS complex (asterisk) and accumulation of MIC60 and MIC19 in a smaller ~500 kDa sub-complex (two asterisks). (**D**) Immunoblot analysis of MICOS subunits. (**E**) Densitometry analysis was performed using ImageJ. (**F**) qPCR analysis. (**G**) Endogenous MIC60 was immunopurified from crude

*Figure 5. continued on next page*

*Figure 5. Continued*

mitochondria isolated from HCT116 cells stably expressing FF2 or QIL1 shRNA. Endogenous interactions with MIC19, MIC25, SAMM50, MIC26, MIC10, MIC27, AFG3L2, or QIL1 were assessed. (**H**) Densitometry analysis was performed using ImageJ.

in response to QIL1 depletion were post-transcriptional, as determined by qPCR (*Figure 5F*). Thus, QIL1 appears to be important for maintaining the levels of several MICOS complex subunits.

To further strengthen our conclusions regarding MICOS complex formation, we immunopurified endogenous MIC60 from HCT116 cells stably expressing FF2 control or QIL1 shRNAs (*Figure 5G*). While MIC60 efficiently bound to MIC19 and MIC25 in QIL1-depleted mitochondria, the abundance of MIC10, MIC26, and MIC27 was significantly reduced in these immune complexes, confirming the existence of a stable sub-complex containing MIC60, MIC19, and MIC25 that lacks MIC10, MIC26, and MIC27 in QIL1-depleted cells (*Figure 5G,H*). Interestingly, MIC60 retained its ability to interact with the OM protein SAMM50 (*Figure 5G,H*), indicating that the MIC60 sub-complex could act in association with the OM independently of MIC10, MIC26, and MIC27.

Importantly, ectopic QIL1 expression rescued cristae morphology defects (*Figure 6A–E*), MICOS disassembly (*Figure 6F*) and downregulation of MIC10 and MIC27 protein levels (*Figure 6G*) that resulted from silencing QIL1 with an shRNA targeting the 3′ untranslated region. Collectively, these data attest to the specificity of the observed loss of function phenotype.

## QIL1 is required for the incorporation of MIC10 into the MICOS complex

Our quantitative proteomics and BN-PAGE data suggested that QIL1 depletion resulted in a reduction in the abundance of the ~700 kDa mature MICOS complex, as revealed by examining core subunits. Moreover, we observed a concomitant accumulation of MIC60, MIC19, and MIC25 in a smaller sub-complex of ~500 kDa. These data agree with the presence of both overexpressed and endogenous MIC60, MIC19, and MIC25 in this sub-complex (*Figure 2D–E*). We hypothesized that QIL1 promotes the maturation of the MICOS complex by acting as a scaffold at the IM, bridging a sub-complex containing MIC60, MIC19, and MIC25 to other subunits including MIC10, MIC26, and MIC27. We hypothesized that when QIL1 is depleted, MIC10, MIC26, and MIC27 fail to integrate into the complex and are degraded. To test this, we asked whether overexpressed MIC10 could physically interact with other MICOS subunits and be incorporated into the mature MICOS complex in cells where QIL1 had been silenced. Thus, we transiently expressed C-terminally tagged MIC10 in cells expressing FF2 shRNA or a shRNA-targeting QIL1 and assessed the presence of the overexpressed protein in the mature MICOS complex at ~700 kDa by BN-PAGE followed by immunoblotting (*Figure 7A*). C-terminally tagged MIC10 was more efficiently assembled into the MICOS complex in control cells compared to those lacking QIL1 even though MIC10 was expressed at similar levels (*Figure 7A–B*) and was properly imported into mitochondria (*Figure 7B–C*) in both, control and QIL1-depleted cell lines. In agreement with these findings, MIC10 binding to MIC60, MIC19, and SAMM50 was significantly reduced in cells expressing QIL1 shRNA (*Figure 7D–E*).

Interestingly, we found that MIC10 overexpression promoted an increase in the incorporation of MIC60 and MIC19 in the mature MICOS complex (~700 kDa, asterisk) and a decrease in the smaller sub-complex (~500 kDa, two asterisks) in the presence of QIL1, consistent with an increase in MICOS assembly (*Figure 7A*).

Altered levels of cardiolipin have been shown to promote severe cristae morphological defects that resemble those caused by loss of MICOS (*Xu et al., 2006*; *Acehan et al., 2007*). Thus, we investigated whether QIL1 knockdown had an effect on cardiolipin content by mass spectrometry (*Figure 7—figure supplement 1*). Our results showed no apparent differences in cardiolipin content or species distribution upon QIL1 knockdown, using three different shRNAs. These results exclude the possibility that the effect of QIL1 on MICOS is an indirect mechanism through regulation of cardiolipin levels and reinforces a model in which QIL1 is a structural component of MICOS required for its assembly.

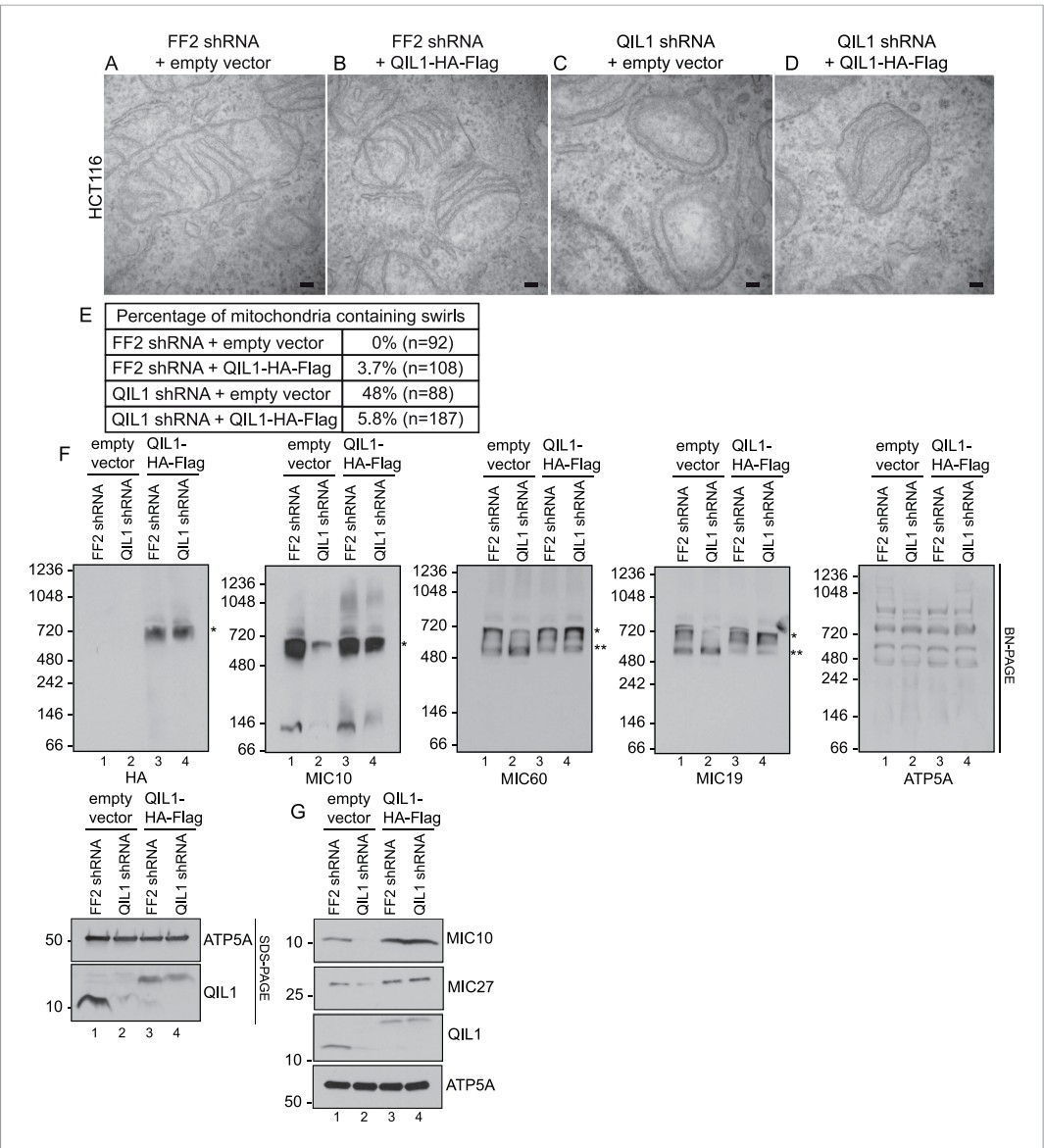

**Figure 6**. Cristae morphology defects, MICOS disassembly and MIC10 and MIC26 degradation can be rescued by QIL1 overexpression. Electron microscopy of HCT116 cells transfected with FF2 shRNA and empty vector (**A**) FF2 shRNA and C-terminally tagged QIL1 (**B**), a QIL1 shRNA targeting the 3' untranslated region and empty vector (**C**) or QIL1 shRNA and C-terminally tagged QIL1 (**D**). Bars, 100 nm. (**E**) Quantification of the number of mitochondria containing membrane swirls based on electron microscopy images. (**F**) BN-PAGE followed by immunotransfer to nitrocellulose membranes. Overexpression of C-terminally tagged QIL1 in cells expressing a QIL1 shRNA targeting the 3' untranslated region restored the levels of MIC10, MIC19, and MIC60 in the mature ~700 kDa MICOS complex and reversed the accumulation of MIC19 and MIC60 in the ~500 kDa assembly intermediate sub-complex (two asterisks). It also restored MIC10 and MIC26 protein levels as observed in immunoblot analysis of total cell lysates (**G**). DOI: 10.7554/eLife.06265.010

## Discussion

In this work, we report a systematic proteomic analysis of the MICOS complex and identify QIL1 as a new subunit required for CJ formation, MICOS complex maturation, and mitochondrial homeostasis. QIL1 associates with the core IM MICOS complex, the cardiolipin binding subunits (MIC26 and MIC27), as well as the SAMM50-MTX1-MTX2 OM complex (*Figure 1C*). Moreover, QIL1 is enriched at CJs to an extent similar to that of the MICOS subunit MIC25 and is present in a ~700 kDa complex, as observed by BN-PAGE (*Figure 2*). Thus, QIL1 may be a central component of CJs and the

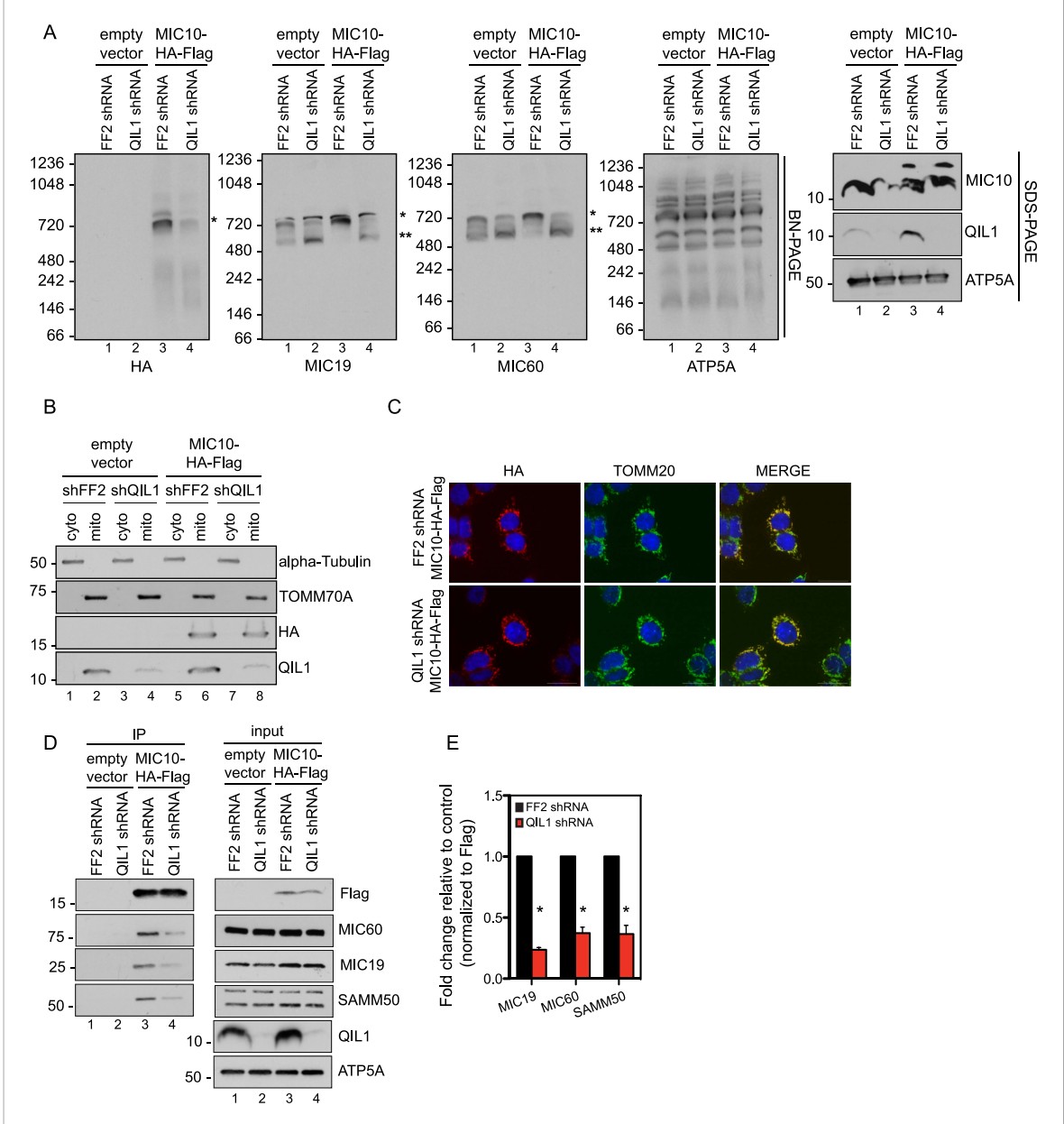

**Figure 7**. QIL1 is required for the binding of MIC10 to the MICOS complex. (**A**) BN-PAGE followed by immunotransfer to nitrocellulose membranes. Incorporation of C-terminally tagged MIC10 into the mature ∼700 kDa MICOS complex (asterisk) was decreased in cells expressing QIL1 shRNA compared to those expressing FF2 shRNA. (**B**) Cytoplasmic and mitochondrial fractions were separated in lysates obtained from HCT116 cells stably expressing FF2 or QIL1 shRNA and transiently expressing empty vector or C-terminally tagged MIC10. (**C**) Immunofluorescence analysis of the subcellular localization of C-terminally tagged MIC10. Bars, 20 μm. (**D**) C-terminally tagged MIC10 was transiently expressed in HCT116 cell lines expressing FF2 shRNA or QIL1 shRNA and immunopurified from mitochondrial lysates. Immunoblot analysis was performed to detect interaction with other MICOS subunits. (**E**) Densitometry analysis was performed using ImageJ. *Figure 7—figure supplement 1* shows the analysis of cardiolipin content and species distribution by LC-MS/MS in mitochondria obtained from cells stably expressing FF2 or 3 different shRNAs targeting QIL1.

The following figure supplement is available for figure 7:

**Figure supplement 1**. Analysis of cardiolipin content and species distribution by LC-MS/MS in mitochondria obtained from cells stably expressing FF2 shRNA or 3 different shRNAs targeting QIL1.

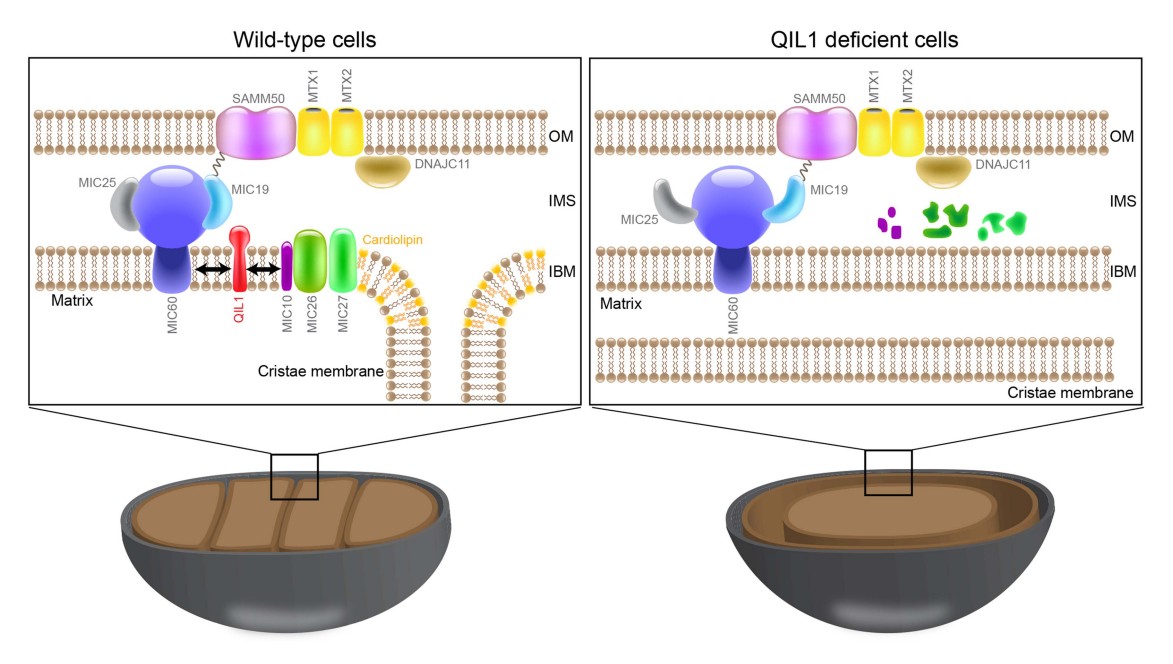

**Figure 8**. QIL1 is a novel MICOS subunit required for MICOS assembly. Model for QIL1 incorporation into the MICOS complex and the effect of loss of QIL1 on the composition of MICOS subunits. We propose that MIC10, MIC26, and MIC27 fail to assemble into a stable MICOS sub-complex containing MIC60, MIC19, and MIC25 and are degraded when QIL1 is depleted leading to MICOS disassembly, loss of CJs and cristae morphology defects.

machinery that connects inner and outer membranes (*Figure 8*). Indeed, depletion of QIL1 leads to MICOS-like defects in cristae morphology, with loss of CJs and formation of swirls of cristae membranes inside mitochondria (*Figure 3A–H*). QIL1 is conserved in *Drosophila* where its depletion also causes mitochondrial phenotypes (*Figure 4*). Moreover, QIL1 knockdown resulted in MICOS disassembly, accumulation of a MIC60-MIC19-MIC25 sub-complex and loss of MIC10, MIC26, and MIC27 from the MICOS complex, as determined by quantitative proteomics and immunoblot analysis (*Figure 5A–D*).

Previous studies indicate that loss of MIC10, MIC26, and MIC27 resulted in cristae remodeling analogous to that found upon QIL1 depletion (*Harner et al., 2011*; *Head et al., 2011*; *Alkhaja et al., 2012*; *Weber, 2013*). The molecular mechanism through which these subunits regulate CJs stability is yet to be determined, but it is possible that the phenotype observed in cells after QIL1 depletion is a result of changes in MIC10, MIC26, or MIC27 protein levels.

The physical properties of cardiolipins are particularly important at areas of the IM with high membrane curvature, such as CJs and at the tip of the cristae (*Ortiz et al., 1999*). In fact, cardiolipins have been shown to be enriched at those regions. In Barth syndrome patients, the levels of the enzyme Tafazzin, required for the remodeling of acyl chains within cardiolipins, are decreased, resulting in reduced cardiolipin levels. This leads to major alterations of mitochondrial membranes morphology and curvature, with the appearance of collapsed cristae packaged in multiple concentric layers (*Acehan et al., 2007*). Interestingly, Tafazzin mutations in flies generated a Barth syndrome-related phenotype characterized by decreased cardiolipin levels, mitochondria showing cristae membrane swirls and motor defects (*Xu et al., 2006*).

Tafazzin has been shown to be regulated by the MICOS interactor Aim24 in yeast (*Harner et al., 2014*) and MIC27 in mammals (*Weber, 2013*) has been shown to bind cardiolipin. Due to its ability to bind cardiolipin, the presence of MIC27 at CJs is particularly important (*Weber, 2013*). Thus, alterations in mitochondrial cardiolipin levels after QIL1 depletion could potentially offer an alternative, indirect explanation for the observed phenotypes. To test the possibility that QIL1 regulates mitochondrial cardiolipin levels, we measured cardiolipin content and species distribution in QIL1-depleted cells compared to control cells expressing FF2 shRNA. We observed that the cardiolipin profiles of cells expressing FF2 or three different QIL1 shRNAs were similar (*Figure 7—figure supplement 1*), excluding

an indirect effect of QIL1 through regulation of cardiolipin levels. This observation is in agreement with data published previously, showing that cardiolipin composition was not substantially changed by the lack of the central MICOS subunits MIC60 or MIC10 (*Bohnert et al., 2012*). These data support a more direct structural role for QIL1 in MICOS stability.

Our results demonstrate that MIC60 forms a stable MICOS sub-complex with MIC19 and MIC25 (*Figure2D–E*) and retains the ability to interact with the OM component SAMM50, independently of MIC10, MIC26, and MIC27 (*Figure 5G,H*). These data suggest that MIC60-MIC19-MIC25 could perform other functions in association with the OM, independently of MIC10, MIC26, and MIC27. This is not surprising given that it has been shown previously that a fraction of MIC60 molecules, but not MIC10, functions in biogenesis of β-barrel proteins of the OM (*Bohnert et al., 2012*) independently of other subunits. Restoration of MIC10 levels by overexpression in cells depleted for QIL1 failed to significantly rescue its interaction with other MICOS subunits (*Figure 7D–E*) and incorporation into the ∼700 kDa complex (*Figure 7A*) as compared to wild-type cells even though MIC10 was efficiently imported into mitochondria and expressed at similar levels in both conditions (*Figure 7B–C*). This indicates that QIL1 is required for the binding of MIC10 to the MIC60-MIC19-MIC25 sub-complex. Thus, we propose a stratified model of MICOS assembly, where the stable MIC60-MIC19-MIC25 sub-complex at the IM binds to MIC10, MIC26, and MIC27 to generate a mature MICOS complex in a step facilitated by QIL1 (*Figure 8*).

Further studies are required to fully understand the regulation of CJ assembly and maintenance. Interestingly, in flies exposed to hyperoxia to induce oxygen stress, the cristae lose their orderly structure, generating a swirl within the muscle mitochondrion and the IM forms concentric stacks inside the matrix (*Walker and Benzer, 2004*), reminiscent of the phenotype observed after MICOS depletion. Intriguingly, the number of mitochondria containing swirls increases slowly with aging (*Walker and Benzer, 2004*). In contrast, giant mitochondria with honeycomb-like cristae were observed in animals exposed to hypoxia (*Lorente et al., 2002*; *Perkins and Hsiao, 2012*). These findings demonstrate the importance of cristae remodeling as a mechanism of adaptation to different metabolic and pathological states in vivo. It still remains to be addressed whether regulation of the MICOS complex takes place in these scenarios.

The MICOS protein interaction network described here provides a resource for additional studies into the regulation of CJ assembly. Indeed, TMEM11 appears to be a novel interactor of the MICOS complex. We identified TMEM11 in association with MIC19, MIC27, MIC60, and MTX2, and endogenous MIC60 associated with endogenous TMEM11 by co-immunoprecipitation (*Figure 1B,F*). TMEM11 contains 2 transmembrane domains. In *Drosophila*, TMEM11 was shown to localize to the IM and regulate cristae morphology, length, and biogenesis (*Rival et al., 2011*; *Macchi et al., 2013*), but the mechanism through which it was involved remained unknown. Our results suggest that it contributes to the MICOS complex. Additional novel interacting proteins identified here (*Figure 1B*) may also be involved in MICOS function.

Taken together, our work provides a comprehensive overview of the MICOS complex interactome and defines a role for the novel IM component QIL1 in MICOS assembly.

## Materials and methods

### Cell culture
HCT116, HeLa, HEK293T cells were grown in Dulbecco's modified Eagle's medium (DMEM) supplemented with 10% fetal calf serum and maintained in a 5% $CO_2$ incubator at 37°C.

### *Drosophila* stocks and culture
*Drosophila* stocks were maintained at room temperature on a standard cornmeal agar diet. UAS-ControlRNAi; UAS-QIL1RNAi, and mef2-GAL4 lines were obtained from the Bloomington stock center. QIL1 knockdown was achieved by crossing virgin mef2-GAL4 females with UAS-QIL1RNAi males. As a control virgin mef2-GAL4 females were crossed with UAS-ControlRNAi males. All experiments were conducted at 25°C.

### Cell line generation, plasmid and siRNA transfections
For plasmid transfection, Lipofectamine 2000 (Invitrogen, Carlsbad, USA) was used for transfection of HeLa cells and TransIT-293 (Mirus, Madison, USA) or Lipofectamine 2000 (Invitrogen) was used for transfection of HEK293T cells according to manufacturer's specifications. Transfected cells were

harvested ~48 hr post-transfection for further analysis. Viral particles were generated in HEK293T cells through the transfection of a pHAGE lentiviral vector (*Murphy et al., 2006*), containing the gene of interest with a C-terminal HA-Flag tag and four helper vectors (VSVG, Tat1b, Mgpm2, and CMV-Rev) (*Wilson, 2008*). Virus-containing supernatants were used to infect HeLa, HCT116, or HEK293T cells. Cells stably expressing the tagged proteins were selected with Puromycin (Invitrogen) for at least one week. For the generation of stable knock-down cell lines, we used miR-30-based shRNA constructs and VSVG and gag-pol helper vectors. Cells were selected for at least one week in Puromycin. For siRNA transfection, Lipofectamine RNAiMAX was used to transfect 20 nM of indicated siRNA into indicated cell lines. Transfected cells were analyzed ~48 hr after transfection. The siRNA and shRNA sequences used in the study were: QIL1 siRNA_1: 5'-CCAAGGAGGGCUGGGAGUA-3'; QIL1 siRNA_2: 5'-GAUGUCAGCUCUGUCGGUG-3'; QIL1 shRNA_1: GCAGGGCTGCCACTGACCTGAA; QIL1 shRNA_2: CCGGCCTTGCCGGCCCAATAAA; QIL1 shRNA_3: GGCCCAATAAAGGACTTCAGAA.

## Immunoprecipitation and proteomic analysis

Cells were lysed in lysis buffer (50 mM Tris–HCl [pH 7.5], 150 mM NaCl, 1% Digitonin [Calbiochem, Billerica, USA], and supplemented with protease inhibitors [Roche, Basel, Switzerland]) for 30 min on ice to obtain whole cell extracts. Lysates were cleared by centrifugation at 4°C for 15 min at maximum speed. Approximately 0.5 mg of mitochondrial lysate was used for endogenous IP. Lysates were incubated with 1 µg of the indicated antibody or control IgG overnight at 4°C. Protein A resin (10 µl) was then added to the IP reaction and incubated further for 2 hr at 4°C. Beads were washed 3 times with lysis buffer. After washing, 2× SDS loading buffer was added and the samples were boiled for 5 min. Samples were separated on a SDS-PAGE gel prior to immunoblot analysis. For proteins with similar molecular weights (e.g., QIL1 and MIC10), separate gels were run for immunoblotting. For IPs of ectopically expressed proteins, agarose beads conjugated with HA or Flag antibodies (Sigma, St. Louis, USA) were used. IP-MS and CompPASS analysis were performed as described previously (*Sowa et al., 2009*; *Behrends et al., 2010*). Briefly, cells (10⁷) were lysed for IP-MS using α-HA beads or α-Flag. After washing, proteins were eluted with HA or Flag peptide, subjected to trichloroacetic acid precipitation, and trypsinized prior to passage through StageTips. Samples were ran in technical duplicate on either an Thermo LTQ mass spectrometer or an LTQ-Orbitrap Elite, and spectra search with Sequest prior to target-decoy peptide filtering, and linear discriminant analysis (*Huttlin et al., 2010*). Protein Assembler was used to convert spectral counts to average peptide spectral matches (APSMs), which takes into account peptides, which match more than one protein in the database. Peptides were identified with a false discovery rate of <1.0%, and the protein false discovery rate was <2.0% (*Figure 1—figure supplement 2*). Peptide data (APSMs) were uploaded into the CompPASS algorithm housed within the CORE environment. For CompPASS analysis of HCT116 cells, we employed a stats table of 214 unrelated bait proteins analyzed in an analogous manner. For analysis of 293T cells, a database of 48 baits was used. The CompPASS system identifies HCIPs based on the WDN-score, which incorporates the frequency with which they identified within the stats table, the abundance (APSMs) when found, and the reproducibility of identification in technical replicates, and also determines a z-score based on APSMs (*Sowa et al., 2009*; *Behrends et al., 2010*). Proteins with WDN-scores >1.0 are considered HCIPs, although we also note that some proteins that may be bona fide-interacting proteins may not reach the strict threshold set by a WDN-score of >1.0 (*Figure 1—figure supplement 2*). Candidate proteins not known to be localized to mitochondria based on Mitocarta (*Pagliarini et al., 2008*) were omitted from the interaction maps.

## Antibodies

Antibodies used in this work include: α-QIL1 (Sigma SAB1102836), α-MINOS1 (Aviva, San Diego, USA, ARP44801-P050), α-CHCHD3 (Aviva ARP57040-P050), α-CHCHD6 (Proteintech, Chicago, USA, 20639-1-AP), α-IMMT (Abcam, Cambridge, UK, ab110329), α-TIMM23 (BD-Biosciences, Franklin Lakes, USA, 611222), α-SAMM50 (Proteintech 20824-1-AP), α-APOOL (Aviva OAAF03292), α-PCNA (Santa Cruz, Dallas, USA, sc-56), α-HSP90 (Epitomics, Burlingame, USA, 3363-1), α-TOMM70A (Epitomics T1677),α-Flag (Sigma SLB6631), α-HA (Covance, Princeton, USA, D13CF00834), α-ATP5A (Abcam ab14748), α-DLD (Santa Cruz sc-271569), α-TMEM11 (Proteintech 16564-1-AP), α-Cytochrome C (Cell Signaling, Danvers, USA, 4272S), α-TOMM20 (Santa Cruz sc-11415), Phalloidin (Invitrogen A22287), and α-APOO (Novus Bio, Littleton, USA, NBP1-28870).

## Mitochondria isolation

Mitochondria were isolated by differential centrifugation. Cells were resuspended in mito-isolation buffer (250 mM sucrose, 1 mM EDTA, 10 mM MOPS-KOH, pH 7.2) and ruptured with 30 s sonication in the lowest setting. The homogenized cellular extract was then centrifuged at 600×g to obtain a post-nuclear supernatant. Mitochondria were pelleted by centrifugation at 8,000×g for 10 min and washed twice in isolation buffer.

## Quantitative proteomics of MICOS assembly

HCT116 stable cell lines expressing FF2 shRNA or QIL1 shRNA were grown in light (K0) or heavy media (K8) and an equal number of cells mixed, prior to purification of mitochondria. Mitochondria were lysed with 1% Digitonin and protein complexes fractionated by blue native-polyacrylamide gel electrophoresis (BN-PAGE). Gel bands were excised and proteins subjected to reduction, alkylation, and trypsinization. Tryptic peptides were analyzed using a Q Exactive mass spectrometer (Thermo Fisher Scientific, San Jose, USA) coupled with a Famos Autosampler (LC Packings) and an Accela600 liquid chromatography (LC) pump (Thermo Fisher Scientific). Peptides were separated on a 100-µm inner diameter microcapillary column packed with ~0.25 cm of Magic C4 resin (5 µm, 100 Å, Michrom Bioresources, Billerica, USA) followed by ~18 cm of Accucore C18 resin (2.6 µm, 150 Å, Thermo Fisher Scientific). For each analysis, we loaded ~1 µg onto the column. Peptides were separated using a 90 gradient of 5–28% acetonitrile in 0.125% formic acid with a flow rate of ~300 nl/min. The scan sequence began with an Orbitrap MS1 spectrum with the following parameters: resolution 70,000, scan range 300–1500 Th, automatic gain control (AGC) target $1 \times 10^6$, maximum injection time 250 ms, and centroid spectrum data type. We selected the top 20 precursors for MS2 analysis which consisted of higher energy collision dissociation (HCD) with the following parameters: resolution 17,500, AGC $1 \times 10^5$, maximum injection time 60 ms, isolation window 2 Th, normalized collision energy 25, and centroid spectrum data type. The underfill ratio was set at 9%, which corresponds to a $1.5 \times 10^5$ intensity threshold. In addition, unassigned and singly charged species were excluded from MS2 analysis and dynamic exclusion was set to automatic. Peptides were identified and quantified using MaxQuant software (Cox and Mann, 2008).

## RNA extraction, reverse transcription, and qPCR

Total RNA was obtained using NucleoSpin RNA II (Macherey–Nagel, Germany) RNA Kit according to manufacturer's protocol. The extracted RNA was then used for reverse transcription using High-Capacity cDNA Reverse Transcription Kit (Applied Biosystems, Waltham, USA). The cDNA obtained was used for qPCR with gene-specific primers and SYBR-green for detection on a LightCycler 480 system (Roche). Primers specific to TUBB were used for normalization. Primer sequences are as follows: TUBB_F: CTGGACCGCATCTCTGTGTA; TUBB_R: CCCAGGTTCTAGATCCACCA; QIL1_F: GCCAG-TACGTGTGTCAGCAG; QIL1_R: GGAGTCACGGATGGGAAAGT; MINOS1_F: CGGATGCGGTCGT-GAAGATA; MINOS1_R: ATCCCATGCCAGAACCGAAG; APOO_F: GCTGGCCTTATTGGACTCCT; APOO_R: ACACGATGGCTTGTTGTGGA; APOOL_F: CACCACCGCTCCAGTCTAAA; APOOL_R: CAGTGCGGATGGAAGCAAAG; ACT5C_F: TACTCTTTCACCACCACCGC; ACT5C_R: GGCCATCTCCTGCTCAAAGT; and CG7603_F: CGCAACCGTCTACTACACACA; CG7603_R: CGTTGTACAGCTTGTCCGTC.

## BN-PAGE

Mitochondria were purified as described above and lysed in 1% Digitonin prior to separation by 4–16% BN-PAGE as previously described (McKenzie et al., 2006). Proteins were transferred to PVDF membrane, and proteins were detected using the indicated antibodies.

## Oxygen consumption

Oxygen consumption rate (OCR) was measured using an XF24 extracellular analyzer (Seahorse Bioscience, North Billerica, USA). HeLa cells transiently transfected with control or QIL1 siRNAs were seeded in 24-well assay plates (BD Bioscience) at a concentration of 20,000 cells per well. After 24 hr, cells were loaded into the instrument for $O_2$ concentration determinations. Cells were sequentially exposed to oligomycin (1 µM), carbonylcyanide p-trifluoromethoxyphenylhydrazone (FCCP; 150 nM), and Antimycin A (10 µM). After each injection, OCR was measured for 3 min, the medium was mixed and again measured for 3 min twice.

## Electron microscopy

HeLA cells transfected with control siRNA or 2 different siRNAs targeting QIL1 and HCT116 stable cell lines expressing FF2 shRNA or 3 independent shRNAs targeting QIL1 were grown to 60% confluency in 6-cm culture dishes and fixed with 1.25% paraformaldehyde, 2.5% glutaraldehyde, 0.03% picric acid followed by osmication and uranyl acetate staining, dehydration in alcohols and embedded in Taab 812 Resin (Marivac Ltd, Nova Scotia, Canada). Sections were cut with Leica ultracut microtome, picked up on formvar/carbon-coated copper slot grids, stained with 0.2% Lead Citrate, and imaged under the Phillips Tecnai BioTwin Spirit transmission electron microscope. For the in vivo analysis, *Drosophila* muscle tissues from third instar larvae were dissected in $Ca^{2+}$ free buffer, fixed overnight at 4°C and processed as described above.

## Immunogold labeling

Cells were fixed in 4% paraformaldehyde and 0.1% glutaraldehyde in 0.1 M Sodium Phosphate buffer, pH 7.4 for 2 hr at room temperature. The cells were subsequently washed in PBS, infiltrated with 2.3 M sucrose in PBS containing 0.2 M glycine. Frozen samples were sectioned at −120°C, and the sections were transferred to formvar carbon-coated copper grids. Immunogold labeling was subsequently carried out at room temperature. Both α-HA antibody and protein A gold were diluted in 1% BSA in PBS. The diluted antibody solution was centrifuged 1 min at 14,000 rpm prior to labeling to avoid possible aggregates. Grids were floated on drops of 1% BSA for 10 min to block for unspecific labeling, transferred to 5 µl drops of primary antibody and incubated for 30 min. The grids were then washed in 4 drops of PBS for a total of 15 min, transferred to 5 µl drops of Protein-A gold for 20 min, washed in 4 drops of PBS for 15 min and 6 drops of double distilled water. Contrasting/embedding of the labeled grids was carried out on ice in 0.3% uranyl acetete in 2% methyl cellulose for 10 min. The grids were examined in a JEOL 1200EX Trans or a TecnaiG Spirit BioTWIN mission electron microscope and images were recorded with an AMT 2k CCD camera.

## Carbonate extraction, osmotic shock, and proteinase K treatment

Mitochondria were isolated and subjected to alkaline extraction in freshly prepared 0.1 M $Na_2CO_3$ (pH 11, 11.5, or 12). Membranes were pelleted at 100,000×*g* for 30 min at 4°C, and supernatants were precipitated with the addition of 1/5 volume of 72% trichloroacetic acid and washed 4 times with cold acetone. After treatments, soluble (S, supernatant) and insoluble (P, pellet) fractions were subjected to SDS-PAGE and western blotting analysis using antibodies against TOMM70A, TIMM23, Cytochrome C, DLD, CHCHD3, and QIL1. Isolated mitochondria from HCT116 cells were subjected to proteinase K (50 µg/ml) proteolysis to digest exposed proteins. Osmotic shock (25 mM sucrose, 10 mM MOPS-KOH, pH 7.2) was used to disrupt the outer mitochondrial membrane. After treatments as indicated, Proteinase K activity was blocked with PMSF (2 mM) and fractions were subjected to SDS-PAGE and western blotting analysis using antibodies against TOMM70A, TIMM23, DLD, Cytochrome C, and QIL1.

## Immunofluorescence

Cells grown on 15-mm glass coverslips or 384-well clear bottom plates were fixed with PBS, 4% PFA, and 4% Sucrose for 10 min, permeabilized with PBS 0.1% Triton X-100 for 3 min, blocked with PBS 1% BSA for 30 min, and incubated with α-TOMM20 (1:100; Santa Cruz) and α-HA (1:200; Covance) in PBS 1% BSA for one hour at room temperature. After extensive washing, fixed cells were incubated with Alexa488-chicken anti-rabbit and Alexa594-goat anti-mouse (1:10000) secondary antibodies for 1 hr at room temperature. For the in vivo analysis, *Drosophila* muscle tissue was dissected in PBS, fixed for 30 min in 4% paraformaldehyde, washed, permeabilized for 2 hr in 0.5% Triton X-100 in PBS, and blocked with normal goat serum in 0.1% Tween in PBS (PBST). α-ATP5A primary antibody (1:300; Abcam) was diluted in blocking buffer and incubated overnight at 4°C. After extensive washing with PBST, tissues were incubated with the secondary antibody (Alexa488-chicken anti-rabbit, 1:10000) and Phalloidin (1: 10000, Invitrogen) for 2 hr. Tissues were washed and mounted in SlowFade Gold antifade reagent (Invitrogen). Images were acquired with a Nikon confocal microscope with a 100× oil objective.

## Quantification of mitochondrial network fragmentation

We used the Imaris software to create 3D reconstructions of *Drosophila* muscle mitochondria from z-stack images. Mitochondrial length and sphericity were determined using the Filament and Surface applications of the software.

## Statistical analysis

Statistical significance was calculated using the paired t-test analysis; p-values <0.05 were considered significant. Asterisks represent p values <0.05. Error bars ($\pm$ s.e.m) show the mean of 3 or 4 biological replicates.

## Lipidomics

Cardiolipin was extracted from mitochondrial extracts according to 50 µg of protein by chloroform/methanol (2:1) extraction. Dried lipid extract from cholroform/methanol extractions was dissolved in cholorolform/methanol (2:1) and injected into the HPLC for analysis. Lipid extracts were separated on an Accucore C18 column (2.1 $\times$ 150 mm, 2.6 µm; Thermo) connected to an Ultimate 3000 HPLC (Thermo). A binary solvent system was used (mobile phase A: 50:50 ACN/H$_2$0, 10 mM NH$_4$HCO$_2$, 0.1% HCO$_2$H; mobile phase B: 10:88:2 AcN/IPA/H$_2$O, 2 mM NH$_4$HCO$_2$, 0.02% HCO$_2$H) in a 28 min gradient at a flow rate of 400 µl/min with the column temperature kept constant at 35°C. The HPLC was connected on-line to a Q-exactive mass spectrometer equipped with an electrospray ionization source (ESI; Thermo). Mass spectra were acquired in a data-dependent mode to automatically switch between full scan MS and up to 15 data-dependent MS/MS scans. The maximum injection time for full scans was 75 ms, with a target value of 1,000,000 at a resolution of 70,000 at m/z 200 and a mass range of 150–1800 m/z in negative mode. The 15 most intense ions from the survey scan were selected and fragmented with HCD with stepped normalized collision energies of 30, 60, and 100. Target values for MS/MS were set at 100,000 with a maximum injection time of 120 ms at a resolution of 17,500 at m/z 200. To avoid repetitive sequencing, the dynamic exclusion of sequenced peptides was set at 10 s. Peaks were analyzed using the Lipid Search algorithm (MKI, Tokyo, Japan). Peaks were defined through raw files, product ion and precursor ion accurate masses. Candidate molecular species were identified by database (>1,000,000 entries) search of negative ion adducts. Mass tolerance was set to 5 ppm for the precursor mass. Samples were aligned within a time window and results combined in a single report. An internal standard for cardiolipin (CL 14:1/14:1/14:1/15:1) spiked in prior to extraction was used for normalization and calculation of the amounts of lipids in pmol/mg protein.

## Acknowledgements

This work was supported by NIH grants GM095567 and NS083524 to JWH and NS069695 to DVV NIH DK098285 to JAP. We thank the Nikon Imaging Center, the Conventional Electron Microscopy facility at Harvard Medical School and the IDAC Facility at Harvard Medical School for assistance with microscopy and data analysis, Tobias Walther for assistance with the lipidomics analysis, and Marcia Haigis (Harvard Medical School) for access to the Seahorse instrument.

## Additional information

### Funding

| Funder | Grant reference | Author |
| --- | --- | --- |
| National Institutes of Health (NIH) | GM095567 | J Wade Harper |
| National Institutes of Health (NIH) | NS083524 | J Wade Harper |
| National Institutes of Health (NIH) | NS069695 | David Van Vactor |
| National Institutes of Health (NIH) | DK098285 | Joao A Paulo |

The funder had no role in study design, data collection and interpretation, or the decision to submit the work for publication.

### Author contributions

VG, EMMN, JAP, FF, Conception and design, Acquisition of data, Analysis and interpretation of data, Drafting or revising the article; ELH, DVV, JWH, Conception and design, Analysis and interpretation of data, Drafting or revising the article; SPG, Conception and design, Drafting or revising the article

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
