## [Decision Letter]

Thank you for sending your work entitled “QIL1 is a Novel Mitochondrial Protein Required for MICOS Complex Stability and Cristae Morphology” for consideration at *eLife*. Your article has been favorably evaluated by Randy Schekman (Senior editor) and three reviewers, one of whom is a member of our Board of Reviewing Editors. Jared Rutter, peer reviewer #3, has agreed to share his identity.

The Reviewing editor and the other reviewers discussed their comments before we reached this decision, and the Reviewing editor has assembled the following comments to help you prepare a revised submission.

From the comments below and the discussion of the reviewers, the consensus is that the manuscript has the potential to represent a significant advance of our understanding of the mechanism of MICOS complex assembly, which would be of great interest to the cell biology community. In addition, the reviewers agree that data convincingly demonstrate that QIL1 is a MICOS subunit, albeit reviewer 2 requests a negative control for the proteomic analysis that should be performed. However, at present the observations do not allow for a clear conclusion regarding the molecular role of this newly identified MICOS subunit. Additional experiments directed at more fully elucidating the function of QIL1 are needed. In particular, the authors need to determine whether QIL1 plays a direct role in MICOS assembly and/or a more indirect role, for example, in cardiolipin synthesis.

Reviewer #1:

The authors report the identification of QIL as a new constituent of MICOS using a thorough proteomic analysis, which will be of great interest to the field. Analysis of cells depleted of QIL and in bodywall muscle cells in an RNAi QIL *Drosophila* model provides good evidence that QIL shares functions reported for other MICOS subunits in the maintenance of inner membrane structure. In addition, proteomic analysis and data from native gels indicate that QIL is required for the stability of the MICOS complex, specifically of MIC10, MIC26 and MIC27. Data presented also indicate the presence of a MIC60, MIC19 and MIC25 subcomplex in QIL depleted cells. Overexpression of MIC10 in cells depleted for QIL failed to significantly restore its interaction with other MICOS subunits as compared to wild type cells expressing similar levels of MIC10. The authors interpret this negative observation to suggest that QIL is required for MIC10 to be incorporated into MICOS holocomplex. In general, the experiments are well controlled and the data quality is very high.

Comments:

The authors should at a minimum examine the localization of overexpressed MIC10 in control and QIL depleted cells. The finding that MIC10 overexpression does not restore its interaction in QIL depleted cells is essentially a negative result that should be interpreted with caution. The authors’ conclusion from this observation in the Abstract and manuscript is that “QIL1 is required for binding of MIC10 to other subunits of the MICOS complex and for its incorporation into the complex”. Other possibilities are possible and thus this should be softened accordingly. What QIL depletion demonstrates convincingly is that there is a stable MICOS subcomplex composed of MIC60, MIC19 and MIC25. The authors should also determine the status of the putative MICOS MIC60, MIC19 and MIC25 subcomplex in cells depleted for QIL1 to strengthen their conclusions regarding MICOS complex assembly.

Reviewer #2:

The manuscript “QIL1 is a novel mitochondrial protein required for MICOS complex stability and cristae morphology” by Guarani et al. reports on a proteomic analysis of the human MICOS complex. With this approach they confirm known components of MICOS and identify two potential novel constituents, QIL1 and TMEM11. The authors characterize QIL1 as mitochondrial protein localized at cristae junctions and support association with MICOS subunits, apolipoproteins, and the outer membrane interactors of MICOS. Depletion of QIL1 leads to a MICOS-like phenotype in human and in *Drosophila* cells. Absence of QIL1 leads to drastically reduced amounts of the MICOS proteins MIC10, MIC26 and MIC27 and destabilize a 700kDa MICOS subcomplex.

The data presented support an association of QIL1 with MICOS. However, it remains unclear if the defect on cristae morphology is caused through loss of QIL1 or the concomitant loss of MIC10, MIC26 and MIC27. Moreover, mechanistic conclusions on the function of QIL1 cannot easily be drawn. It remains to be shown if QIL1 is involved in assembly or stability of MICOS or if it has a role in cardiolipin synthesis.

1) Figure 1—figure supplement 1: the amount of QIL1 is drastically increased in all tagged strains this could lead to non-physiological organization of MICOS components or QIL1/MICOS association through changes in the equilibrium. This is critical for interpreting the immunoisolation analyses.

2) All immunoisolation analyses lack a negative control. The authors did not monitor for the purity of these analyses by probing for an abundant inner membrane protein.

3) Figure 1: the amount of MIC60 in lanes 2 and 3 does not match to the input.

4) Figure 2: MIC60 show higher complexes (>700kDa) in 2D-PAGE. These complexes are not seen on BN-PAGE with the tagged strain. They are also not visible by BN-PAGE for endogenous MIC60 (Figure 5). This questions the interpretation of the first dimension. Moreover, the 700kDa complex is not the mature MICOS but rather a relatively stable subcomplex, as MICOS has been detected in the MDa range (e.g. [13]).

5) Since MIC60, MIC19 and MIC25 are not affected by QIL1 depletion, the authors should assess if these MICOS subunits still interact with the MISOC outer membrane interactors in the absence of QIL1. This would allow the authors to define if this complex can act independent of the other subunits and provide some more mechanistic insight into MICOS.

6) The authors should use MICOS nomenclature for QIL1.

7) The scheme in Figure 1 could lead to a misunderstanding on the MICOS organization since the central subunit MIC10 is not included. The authors should include MIC10 in the network overview (Figure 1) and extend the validation (Figure 1) with western blot data. Since MIC10 is not detectable by mass spec, this makes me wonder as to how many other proteins are missing.

8) The authors discuss in length about the role of cardiolipin. However, they did not assess cardiolipin levels upon QIL1 knock down. This should be done.

9) The analyses on TMEM11 are rudimentary. This should be extended or taken out.

Reviewer #3:

Using a systematic proteomics approach, the authors report the identification of a previously uncharacterized subunit of the MICOS complex, QIL1. In addition to extensive proteomics, the authors utilize co-immunopreciptitation and BN-PAGE to convincingly demonstrate that QIL1 is a member of the MICOS complex. Importantly, they do not simply suggest that QIL1 is a MICOS subunit but demonstrate that QIL1 is essential for mediating the interaction of a ∼500kDa subcomplex with additional MICOS subunits to form the mature ∼700kDa complex. Furthermore, the authors utilize EM to investigate the morphological consequences of genetic depletion of QIL1 (or its orthologs) in two model systems and nicely demonstrate that loss of this subunit results in mitochondria with abnormal morphology, highlighted by a lack of cristae.

Overall, the paper provides a thorough characterization of QIL1 and its essential role in mediating MICOS complex assembly. It highlights a novel member of the complex and demonstrates that loss of this subunit results in a phenotype consistent with loss of other MICOS subunits. The result is that I am totally convinced that the authors are correct in their conclusions and the only substantial weakness is that they don't go a little farther in the interrogation of the MICOS complex or the QIL1 protein. For instance, QIL1 might play a regulatory role in assembling the mature complex. In fact, based on the data presented in Figure 7 (compared to Figure 6), it looks like MIC10 might be playing a dose-limited regulatory role and QIL1 is required for this activity. This could lead to a greatly enhanced understanding of the regulation and role of the MICOS complex in mitochondrial biology.

In summary, the major highlight of this manuscript is the discovery of a novel subunit of the MICOS complex. Furthermore, the authors provide convincing evidence that QIL1 is required for late-stage maturation of MICOS. Indeed, loss of QIL1 leads to a stalled intermediate complex that is unable to bind additional subunits. In the end, the authors show that QIL1 is required to form the mature MICOS complex. In addition to the wealth of interaction data, the authors also demonstrate that loss of QIL1 has catastrophic consequences on mitochondrial morphology and cristae structure, consistent with loss of other MICOS proteins. This is strengthened by the use of two distinct model systems thereby demonstrating the conserved role of this protein. Overall, the data is convincing and the story is interesting, however, potentially extending the investigation a bit further could substantially raise the impact. In the end, this is a strong paper and should be accepted.

Specific points to consider:

1) The western blot in Figure 2 is difficult to interpret due to the gel distortion. It's hard to even tell which lane is which for MIC10 and QIL1 blots.

2) It would be better to carry out the studies in fly either using a genetic knockout or rescuing the RNAi lines. With that said, these experiments would require significant time and effort and since all finding are consistent with the mammalian data, are probably not necessary.

3) It would be nice to see MIC26 westerns in Figure 5 if the antibody is available.

---

## [Author Response]

*From the comments below and the discussion of the reviewers, the consensus is that the manuscript has the potential to represent a significant advance of our understanding of the mechanism of MICOS complex assembly, which would be of great interest to the cell biology community. In addition, the reviewers agree that data convincingly demonstrate that QIL1 is a MICOS subunit, albeit reviewer 2 requests a negative control for the proteomic analysis that should be performed. However, at present the observations do not allow for a clear conclusion regarding the molecular role of this newly identified MICOS subunit. Additional experiments directed at more fully elucidating the function of QIL1 are needed. In particular, the authors need to determine whether QIL1 plays a direct role in MICOS assembly and/or a more indirect role, for example, in cardiolipin synthesis*.

The central point brought up by the reviewers concerns whether QIL1 is directly involved in MICOS complex assembly or the effect of QIL1 knockdown on MICOS assembly and levels is an indirect mechanism, possibly through the regulation of cardiolipin synthesis. To address this question, we collaborated with Florian Fröhlich (an expert in lipidomics in Tobias Walther’s lab) to measure cardiolipin levels in mitochondria in a direct manner by mass spectrometry. Our analysis revealed that cardiolipin levels and species distribution were unchanged upon QIL1 stable knockdown using 3 different shRNAs (Figure 7—figure supplement 1). This data supports a more direct structural role for QIL1 in MICOS stability, suggesting a scenario that would be more consistent with QIL1 being a component of the MICOS complex rather than being indirectly involved, for example, in promoting cardiolipin production. These observations are in agreement with data published previously showing that cardiolipin profiles were unaffected upon depletion of the MICOS subunits MIC10 and MIC60 (5). The idea that QIL1 is an actual component of the MICOS is further supported by additional data showing that:

1) QIL1 is a component of the major ∼700 KDa complex seen in native gels (Figure 2).

2) It interacts with 5 known subunits of MICOS and known MICOS binding proteins at the outer membrane (Figure 1).

3) It is enriched at cristae junctions to a similar extent as a bona-fide MICOS subunit (Figure 2).

4) QIL1 knockdown impaired MICOS assembly, resulting in the formation of a stable MICOS sub-complex that contains MIC60, MIC19 and MIC25 and lacks MIC10, MIC26 and MIC27 as demonstrated by native gels and quantitative proteomic analysis (Figure 5).

To further strengthen these observations, we have immunopurified the endogenous MICOS complex with an antibody targeting MIC60 and investigated the status of the complex by probing for the presence of other MICOS subunits by western blot analysis (Figure 5). These results have confirmed the existence of sub-complexes containing MIC60, MIC19 and MIC25 when QIL1 levels are silenced. Moreover, as requested by one reviewer, we have shown that MIC60 maintains its ability to interact with the outer membrane protein SAMM50 when QIL1 expression is silenced. This finding is in line with previous work showing that MIC60 performs additional functions, independently of other MICOS subunits (5; 43). Furthermore, the interaction between MIC60 and three additional components—MIC10, MIC26 and MIC27—is significantly reduced upon QIL1 knockdown, reinforcing the idea that these subunits require QIL1 for their efficient assembly into the MICOS complex and for MICOS maturation.

Reviewer #1:

*The authors should at a minimum examine the localization of overexpressed MIC10 in control and QIL depleted cells. The finding that MIC10 overexpression does not restore its interaction in QIL depleted cells is essentially a negative result that should be interpreted with caution. The authors’ conclusion from this observation in the Abstract and manuscript is that* “*QIL1 is required for binding of MIC10 to other subunits of the MICOS complex and for its incorporation into the complex*”*. Other possibilities are possible and thus this should be softened accordingly*.

We agree with the reviewer that the localization of overexpressed MIC10 in control and QIL1 depleted cells is critical to the interpretation of the results shown in Figure 7. We have repeated the experiment shown in Figure 7 of the revised manuscript and using immunofluorescence and cellular fractionation followed by western blot analysis (Figure 7), we found that ectopic MIC10-HA still localizes to mitochondria, consistent with the idea that it is correctly targeted. Based on the fraction of MIC10 immunoreactive protein seen in SDS-PAGE, the ectopically expressed protein is present at about 50% of the endogenous protein. Under these conditions (i.e. higher MIC10 levels), the abundance of the mature ∼700 KDa MICOS complex in native gels wasn’t rescued. We conclude that while MIC10 is present in mitochondria, it is unable to assemble into MICOS when QIL1 is reduced by RNAi. We thank the reviewer for suggesting that we improve this experiment.

We have rephrased our conclusion that now reads: “In QIL1-depleted cells, overexpressed MIC10 fails to significantly restore its interaction with other MICOS subunits and with SAMM50”.

*What QIL depletion demonstrates convincingly is that there is a stable MICOS subcomplex composed of MIC60, MIC19 and MIC25. The authors should also determine the status of the putative MICOS MIC60, MIC19 and MIC25 subcomplex in cells depleted for QIL1 to strengthen their conclusions regarding MICOS complex assembly*.

In order to strengthen our conclusions regarding MICOS complex assembly defects upon QIL1 knockdown, we have assessed the status of the MICOS assembly intermediate by IP-western blot analysis. To do so, we have immunopurified endogenous MIC60 from cells stably expressing FF2 or QIL1 shRNAs and analyzed its interactions with all other MICOS subunits (Figure 5). Our results demonstrated that MIC60 binding to MIC19 and MIC25 remained unchanged upon QIL1 knockdown, confirming the existence of a stable MIC60-MIC19-MIC25 sub-complex. However, MIC10, MIC26 and MIC27 levels were strongly reduced in MICOS complexes from QIL1 depleted mitochondria, even though the levels of immunoprecipitated MIC60 were similar in both conditions. This experiment was performed in biological triplicates and densitometry analysis was performed using ImageJ. In each experiment, the amounts of each MICOS subunit was normalized to MIC60 levels in the immune complexes and the statistical analysis is shown in Figure 5.

Reviewer #2:

*The data presented support an association of QIL1 with MICOS. However, it remains unclear if the defect on cristae morphology is caused through loss of QIL1 or the concomitant loss of MIC10, MIC26 and MIC27. Moreover, mechanistic conclusions on the function of QIL1 cannot easily be drawn. It remains to be shown if QIL1 is involved in assembly or stability of MICOS or if it has a role in cardiolipin synthesis*.

*1)*
Figure 1—figure supplement 1*: the amount of QIL1 is drastically increased in all tagged strains this could lead to non-physiological organization of MICOS components or QIL1/MICOS association through changes in the equilibrium. This is critical for interpreting the immunoisolation analyses*.

We thank the reviewer for drawing our attention to this point. This represents a misunderstanding of the data caused by the fact that we did not include a control cell extract in the western blot. We apologize. Figure 1—figure supplement 1 contained QIL1 western blots for cells expressing either QIL1-HA-FLAG or one of several other MICOS subunits. The reviewer commented that the levels of QIL1 were much higher in the cells expressing MICOS but the comparison being made was with the QIL1-HA-FLAG cell line, not control 293T cells. It turns out that when compared to control cell lines, the levels of endogenous QIL1 in the MICOS subunit expressing cells are identical to 293T cells expressing the empty vector with the HA-Flag tag (as shown in the new Figure 1—figure supplement 1). In cells expressing QIL1-HA-FLAG, there is a reduction in the abundance of the endogenous QIL1, likely reflecting the fact that there is a homeostatic mechanism to maintain total QIL1 levels within a particular range. The band for the HA-FLAG tagged QIL1 is approximately equal to the levels of endogenous QIL1 and together, they are similar to the levels of endogenous QIL1 in the control cell line. We updated the figure (now Figure 1—figure supplement 1) with a western that also includes the control 293T cells and will describe this better in the text.

*2) All immunoisolation analyses lack a negative control. The authors did not monitor for the purity of these analyses by probing for an abundant inner membrane protein*.

We repeated all endogenous IP-western blots probing for an inner membrane protein (AFG3L2) to address this comment. AFG3L2 did not associate with any of the proteins analyzed in this paper.

For the IP of HA-Flag tagged proteins followed by mass spectrometry, CompPASS analysis filters non-specific binding to the beads or to the HA-Flag antibodies by taking into account the frequency, abundance and reproducibility with which a protein appears in each IP measured across dozens of unrelated proteins. This method is described in [40]. To confirm the efficacy of the method, we have performed IP-MS on cells expressing HA-Flag tagged GFP and no mitochondrial proteins were identified as high confidence interactors (HCIPs) (Figure 1—figure supplement 2). In addition to GFP, only 3 proteins out of a total of 153 detected by mass spectrometry passed the thresholds for a candidate interactor (RNF13 with 2 ASPMs, PPP1R9B with 2 ASPMs, HSPA1L with 36 ASPMs) and none of these were mitochondrial proteins.

*3)*
Figure 1*: the amount of MIC60 in lanes 2 and 3 does not match to the input*.

We have repeated this experiment loading the entire reaction in the gels for WB analysis and updated Figure 1. The levels now match.

*4)*
Figure 2*: MIC60 show higher complexes (>700kDa) in 2D-PAGE. These complexes are not seen on BN-PAGE with the tagged strain. They are also not visible by BN-PAGE for endogenous MIC60 (*Figure 5*). This questions the interpretation of the first dimension. Moreover, the 700kDa complex is not the mature MICOS but rather a relatively stable subcomplex, as MICOS has been detected in the MDa range (e.g.*
[13]*)*.

We do not discount the possibility that there could be larger complexes at lower abundance. We do see evidence for larger complexes as smeared in the 1 MDa range (see for example Figure 2) and this was also validated by our quantitative proteomics experiments examining native gels (Figure 5). In our hands, the ∼700 KDa complex is the most obvious MICOS complex in human cells, and is routinely identified with several MICOS subunit antibodies in native gels. Our ability to detect these smears (possibly a form of a “super complex”) may vary depending on the detergent used or the sensitivity of the antibody being used to identify MICOS complex subunits in first dimension native gels. SDS-PAGE analysis also allows for better detection of epitopes otherwise hidden in the native conformation inside certain supramolecular complexes. Moreover, the identity of the proteins present in these larger complexes remains unknown and it is possible that they include additional components not in the classical MICOS complexes. Results from Ott et al. in mammalian cells have shown that, MIC60 and MIC19 were mainly observed at ∼700KDa, together with a smear towards the high molecular weight region of the gel (32), which resemble our observations. While it is more difficult to establish a direct comparison between mammalian and the yeast MICOS complex, Harner et al. have also observed that all MICOS proteins they analyzed were present in two large complexes of >1 MDa and ∼700 KDa in first dimension BN-PAGE (13).

*5) Since MIC60, MIC19 and MIC25 are not affected by QIL1 depletion, the authors should assess if these MICOS subunits still interact with the MISOC outer membrane interactors in the absence of QIL1. This would allow the authors to define if this complex can act independent of the other subunits and provide some more mechanistic insight into MICOS*.

This is a very interesting point raised by the reviewer. We have addressed this question in Figure 5. We have immunopurified endogenous MIC60 from mitochondria obtained from cells stably expressing FF2 or QIL1 shRNAs. When we analyzed binding to endogenous SAMM50, we observed no significant change upon QIL1 knockdown, indicating that the MIC60 sub-complex could act in association with the outer membrane components, independently of MIC10, MIC26 and MIC27. We included this comment in the Discussion section of the paper.

*6) The authors should use MICOS nomenclature for QIL1*.

We had considered giving QIL1 a new name that would be consistent with the MICOS nomenclature but had decided against it given that there is no obvious ortholog in yeast but there is a yeast gene of very similar size (12 KDa, Mic12). QIL1 displays ∼10% sequence identity with yeast Mic12, and we cannot rule out the possibility that QIL1 is the functional ortholog of the yeast Mic12. At the Editor’s discretion, we would be happy to name it MIC13, for example, with the hope that people don’t confuse this with Mic12 in yeast, but in the current version, we have maintained the official gene symbol.

*7) The scheme in*
Figure 1
*could lead to a misunderstanding on the MICOS organization since the central subunit MIC10 is not included. The authors should include MIC10 in the network overview (*Figure 1*) and extend the validation (*Figure 1*) with western blot data. Since MIC10 is not detectable by mass spec, this makes me wonder as to how many other proteins are missing*.

MIC10 was not in the original set of constructs that we had available when we set up the IP-MS experiment. We have now attempted to perform IP-MS analysis on a stable cell line expressing C-terminally tagged MIC10. Even though we can detect MIC10-HA by western blot analysis and confirm its mitochondrial localization by immunofluorescence, we weren’t able to detect peptides for MIC10 by mass spectrometry. Also, we were unable to directly observe MIC10 via IP-MS targeting other known MICOS complex members. This mostly reflects idiosyncrasies of this protein's primary structure that make it uniquely challenging for identification via bottom-up proteomics methods. These techniques rely upon digestion of proteins with trypsin to produce peptides whose chemical and physical properties are amenable for isolation, LC separation, and fragmentation. Although intact human proteins vary in their structural and chemical characteristics, upon digestion the vast majority produces at least several peptides that are suitable for LC-MS detection; the exceptions are small and often hydrophobic proteins. Not only is MIC10 small (78 amino acids), but nearly a quarter of its sequence consists of a transmembrane domain and the C-terminal half that extends into the intermembrane space is quite hydrophobic as well. Furthermore, its eight basic residues (K/R) are unevenly distributed across its length and often cluster together in series, reducing digestion efficiency and resulting in tryptic peptides that are either long or too short to be specific to a single protein. Of the four tryptic peptides whose expected lengths would exceed four amino acids, two are long and excessively hydrophobic: one encompasses essentially the entire transmembrane domain; the other, spanning nearly the entire inter-membrane portion of the protein, contains two tyrosines, one tryptophan, and two phenylalanines along with several aliphatic side chains. The second of these peptides also contains several Met residues whose side chains are prone to oxidation during sample handling, potentially dividing its signal across multiple oxidation states. The remaining peptides are the N-terminal peptide (MSESELGR) and another nearby sequence (CLADAVVK), neither of which are ideal targets for LC-MS detection. Because MIC10 poses so many challenges, it is not surprising that it was not observed directly in our AP-MS experiments. Fortunately, it is not a typical example: more than 98% of human proteins listed in SwissProt are longer than MIC10 and will more likely produce tryptic peptides suitable for LC-MS detection. Thus, we expect that very few other bona fide interactors would evade AP-MS detection in this way.

With respect to our primary data in Figure 1 (interaction map), we prefer to not add MIC10 to it since it only contains proteomic data but to address the reviewers comment concerning the subunits of the known MICOS complex, we have added a new Figure 1—figure supplement 1 that shows the known subunits and MICOS interactors C-terminally tagged for IP-MS analysis when we began our work.

*8) The authors discuss in length about the role of cardiolipin. However, they did not assess cardiolipin levels upon QIL1 knock down. This should be done*.

We have collaborated with Florian Fröhlich (an expert in lipidomics in Tobias Walther’s lab) to directly address this comment by mass spectrometry. Our results showed that QIL1 knockdown did not alter cardiolipin levels or species distribution (this data is in Figure 7—figure supplement 1 and is discussed in the text). This helped us rule out an indirect role for QIL1 in MICOS assembly by altering cardiolipin levels. In line with this finding, work published by the Pfanner and van der Laan groups showed that MIC60 and MIC10 depletion did not affect cardiolipin content (5).

*9) The analyses on TMEM11 are rudimentary. This should be extended or taken out*.

We would prefer to leave this data in, since it is further validation of our interaction data (Figure 1) and also has been previously linked with mitochondrial morphology in *Drosophila*. Otherwise, this will be completely lost making it hard for anyone to follow up on. But if the Editor thinks it best to remove this confirmatory experiment, we can.

Reviewer #3:

*In summary, the major highlight of this manuscript is the discovery of a novel subunit of the MICOS complex. Furthermore, the authors provide convincing evidence that QIL1 is required for late-stage maturation of MICOS. Indeed, loss of QIL1 leads to a stalled intermediate complex that is unable to bind additional subunits. In the end, the authors show that QIL1 is required to form the mature MICOS complex. In addition to the wealth of interaction data, the authors also demonstrate that loss of QIL1 has catastrophic consequences on mitochondrial morphology and cristae structure, consistent with loss of other MICOS proteins. This is strengthened by the use of two distinct model systems thereby demonstrating the conserved role of this protein. Overall, the data is convincing and the story is interesting, however, potentially extending the investigation a bit further could substantially raise the impact. In the end, this is a strong paper and should be accepted*.

We hope that the reviewer will appreciate that we have extended our investigation further. Notably, we have shown that QIL1 does not affect cardiolipin levels or species distribution, supporting a more direct structural role for QIL1 in MICOS complex maturation. Additionally, we have provided further evidence for the existence of a stable sub-complex containing MIC60, MIC19 and MIC25 that lacks MIC10, MIC26 and MIC27 when QIL1 is depleted, further supporting a hierarchical model of MICOS assembly that requires QIL1.

*Specific points to consider*:

*1) The western blot in*
Figure 2
*is difficult to interpret due to the gel distortion. It's hard to even tell which lane is which for MIC10 and QIL1 blots*.

We have repeated and updated Figure 2. We also placed lane numbers under each lane so it is easier to tell where the lanes are. The panels for MIC10 and QIL1 were the same samples ran on different gels. The behavior of these very small proteins in the detergents used are remarkably similar in terms of how the proteins migrate.

*2) It would be better to carry out the studies in fly either using a genetic knockout or rescuing the RNAi lines. With that said, these experiments would require significant time and effort and since all finding are consistent with the mammalian data, are probably not necessary*.

We would prefer to not take on the extra time and expense to do this, especially considering that all the human and fly data are internally consistent.

*3) It would be nice to see MIC26 westerns in*
Figure 5
*if the antibody is available*.

We have analyzed MIC26 levels in lysates from the experiments displayed in Figure 5 and updated Figure 5.